# Restless Bandits with Average Reward: Breaking the Uniform Global Attractor Assumption

**Yige Hong**[1]* **Qiaomin Xie**[2] **Yudong Chen**[2] **Weina Wang**[1]
[1]Carnegie Mellon University    [2]University of Wisconsin-Madison
{yigeh,weinaw}@cs.cmu.edu
{qiaomin.xie,yudong.chen}@wisc.edu

## Abstract

We study the infinite-horizon restless bandit problem with the average reward criterion, in both discrete-time and continuous-time settings. A fundamental goal is to efficiently compute policies that achieve a diminishing optimality gap as the number of arms, $N$, grows large. Existing results on asymptotic optimality all rely on the uniform global attractor property (UGAP), a complex and challenging-to-verify assumption. In this paper, we propose a general, simulation-based framework, `Follow-the-Virtual-Advice`, that converts any single-armed policy into a policy for the original $N$-armed problem. This is done by simulating the single-armed policy on each arm and carefully steering the real state towards the simulated state. Our framework can be instantiated to produce a policy with an $O(1/\sqrt{N})$ optimality gap. In the discrete-time setting, our result holds under a simpler synchronization assumption, which covers some problem instances that violate UGAP. More notably, in the continuous-time setting, we do not require *any* additional assumptions beyond the standard unichain condition. In both settings, our work is the first asymptotic optimality result that does not require UGAP.

## 1 Introduction

The restless bandit (RB) problem is a dynamic decision-making problem that involves a number of Markov decision processes (MDPs) coupled by a constraint. Each MDP, referred to as an arm, has a binary action space, {passive, active}. At every decision epoch, the decision maker is constrained to select a fixed number of arms to activate, with the goal of maximizing the expected reward accrued. The RB problem finds applications across a spectrum of domains, including wireless communication [ALT19], congestion control [AADJ13], queueing models [ABG09], crawling web content [AB19], machine maintenance [GMA05], clinical trials [VBW15], to name a few.

In this paper, we focus on infinite-horizon RBs with the average-reward criterion. Since the exact optimal policy is PSPACE-hard to compute [PT99], it is of great theoretical and practical interest to focus on policies that approximately achieve the optimal value and compute such policies in an efficient matter. The *optimality gap* of a policy is defined as the difference between its average reward per arm and that of an optimal policy. In a typical asymptotic regime where the number of arms, $N$, grows large, we say that a policy is *asymptotically optimal* if its optimality gap is $o(1)$ as $N \to \infty$.

**Prior work and the uniform global attractor property assumption.** Prior work has studied the celebrated Whittle index policy [Whi88] and LP-Priority policies [Ver16] and established sufficient conditions for their asymptotic optimality [WW90, Ver16, GGY20, GGY22]. One key assumption underpinning all prior work is the *uniform global attractor property* (UGAP)—also known as globally asymptotic stability—which pertains to the mean-field/fluid limit of the restless bandit system in the

---

*Corresponding author

37th Conference on Neural Information Processing Systems (NeurIPS 2023).

| | Paper | Policy | Optimality Gap | Conditions[*] |
|---|---|---|---|---|
| Discrete-time setting | [GGY20] | Whittle Index | $O(\exp(-cN))$ | UGAP & Non-singular |
| | [GGY22] | LP-Priority | $O(\exp(-cN))$ | UGAP & Non-degenerate |
| | This paper | `FTVA(`$\bar{\pi}^*$`)` | $O(1/\sqrt{N})$ | SA |
| Continuous-time setting | [WW90] | Whittle Index | $o(1)$ | UGAP |
| | [Ver16] | LP-Priority | $o(1)$ | UGAP |
| | [GGY20] | Whittle Index | $O(\exp(-cN))$ | UGAP & Non-singular |
| | [GGY22] | LP-Priority | $O(\exp(-cN))$ | UGAP & Non-degenerate |
| | This paper | `FTVA-CT(`$\bar{\pi}^*$`)` | $O(1/\sqrt{N})$ | – |

Table 1: Optimality gap results and conditions. [*]All papers require the standard unichain assumption.

asymptotic limit $N \to \infty$. UGAP stipulates that the system's state distribution in the mean-field limit converges to the optimal state distribution attaining the maximum reward, from any initial distribution. It has been well recognized that UGAP is a highly technical assumption and challenging to verify: the primary way to test UGAP is numerical simulations; see [GGY20] for a detailed discussion.

More recent work studies the *rate* at which the optimality gap converges to zero. The work [GGY20] and [GGY22] prove a striking $O(\exp(-cN))$ optimality gap for the Whittle index policy and LP-Priority policies, respectively, where $c$ is a constant. In addition to UGAP, these results require a non-singularity or non-degenerate condition. We are not aware of any rate of convergence result without assuming UGAP or non-singularity/degeneracy. See Table 1 for a summary.

Therefore, prior work on average-reward restlest bandit leaves two fundamental questions open:

1. Is it possible to achieve asymptotic optimality without UGAP?

2. Is is possible to establish a non-trivial convergence rate for the optimality gap in the absence of the non-singular/non-degenerate assumption (and UGAP)?

**Our contributions.** We consider both the discrete-time and continuous-time settings of the average-reward restless bandit problem. We propose a general, simulation-based framework, `Follow-the-Virtual-Advice` (FTVA) and its continuous-time variant `FTVA-CT`, which convert any single-armed policy into a policy for the original $N$-armed problem, with a vanishing performance loss. Our framework can be instantiated to produce a policy with an $O(1/\sqrt{N})$ optimality gap, under the conditions summarized in Table 1, which we elaborate on later.

Under our framework, computing an asymptotically optimal policy is efficient since it reduces to deriving an optimal single-armed policy, whose complexity is independent of $N$. The resultant policy can be implemented with a linear-in-$N$ computational cost and some of its subroutines can be implemented in a distributed fashion over the arms (see Appendix B for more details).

Our results can also be extended to RBs with heterogeneous arms. See Appendix H for details.

We now elaborate on the conditions in Table 1. In the discrete-time setting, our result holds under a condition called the Synchronization Assumption (SA), in addition to the standard unichain assumption required by all prior work. The SA condition, which is imposed on the MDP associated with a single arm, admits several intuitive sufficient conditions. While it is unclear whether SA subsumes UGAP, we show that there exist problem instances that violate UGAP but satisfy SA. Figure 1 shows one such instance (constructed by [GGY20], described in Appendix G for ease of reference), in which the Whittle Index and LP-Priority policies coincide and have a non-diminishing optimality gap, whereas our policy, named as `FTVA(`$\bar{\pi}^*$`)`, is asymptotically optimal. In addition, our $O(1/\sqrt{N})$ bound on the optimality gap does not require a non-singularity/non-degeneracy assumption.

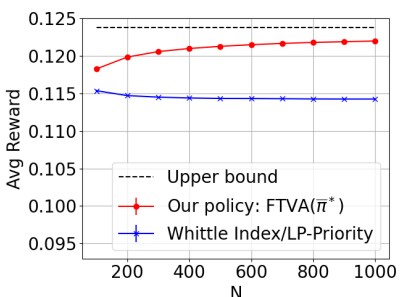

Figure 1: An discrete-time RB problem that satisfies SA but not UGAP.

More notably, in the continuous-time setting, we *completely eliminate the UGAP assumption*. We show that our policy `FTVA-CT`$(\bar{\pi}^*)$ achieves an $O(1/\sqrt{N})$ optimality gap under only the standard unichain assumption (which is required by all prior work).

In both settings, our results are the first asymptotic optimality results that do not require UGAP. We point out that UGAP is considered necessary for LP-Priority policies [Ver16, GGY22]. Our policy, `FTVA`$(\bar{\pi}^*)$, is not a priority policy. As such, we hope our results open up new directions for developing new classes of RB algorithms that achieve asymptotic optimality without relying on UGAP.

**Intuitions.** Our algorithm and many existing approaches solve an LP relaxation of the original problem. The solution of the LP induces a policy and gives an "ideal" distribution for the system state. Existing approaches directly apply the LP policy to the current system state, and then perform a simple rounding of the resulting actions so as to satisfy the budget constraint. When the current system state is far from the ideal distribution, the actual actions after rounding may deviate substantially from the LP solution and thus, in the absence of UGAP, would fail to drive the system to the optimum.

Our `FTVA` framework, in contrast, prioritizes constraint satisfaction. We apply the LP solution to the state of a *simulated* system, which is constructed carefully so that the resulting actions satisfy the constraint after a minimal amount of rounding. Consequently, starting from any initial state, our policy steers the system towards the ideal distribution and hence approximates the optimal value.

This method is inspired by the approach in [HXW23], which also involves a simulated system, but for a stochastic bin-packing problem.

**Other related work.** Another condition extensively discussed in the RB literature is the indexability condition [Whi88], which is necessary for the Whittle index policy to be well-defined, but not required by LP-Priority policies [Ver16]. However, indexability alone does not guarantee the asymptotic optimality of Whittle index.

So far we have discussed prior work on infinite-horizon RBs with average reward. For the related setting of finite-horizon total reward RBs, a line of recent work has established an $O(1/\sqrt{N})$ optimality gap [HF17, ZCJW19, BS20, GNJT23], and an $O(\exp(-cN))$ gap assuming non-degeneracy [ZF21, GGY22]. For the infinite-horizon discounted reward setting, [ZF22, GNJT23] propose policies with $O(1/\sqrt{N})$ optimality gap without assuming indexability and UGAP. While these results are not directly comparable to ours, it is of future interest to see if our simulation-based framework can be applied to their settings. For a more detailed discussion of prior work, see Appendix A.

**Paper Organization.** While our continuous-time result is stronger, the discrete-time setting is more accessible. Therefore, we first discuss the discrete-time setting, which includes the problem statement in Section 2, our proposed framework, `Follow-the-Virtual-Advice`, in Section 3, and our results on the optimality gap in Section 4. Results for the continuous-time setting are presented in Section 5. We conclude the paper in Section 6. Proofs and additional discussion are given in the appendices.

## 2 Problem Setup

Consider the infinite-horizon, discrete-time restless bandit problem with $N$ arms indexed by $[N] \triangleq \{1, 2, \ldots, N\}$. Each arm is associated with an MDP described by the tuple $(\mathbb{S}, \mathbb{A}, P, r)$. Here $\mathbb{S}$ is the state space, assumed to be finite; $\mathbb{A} = \{0, 1\}$ is the action set, and we say the arm is *activated* or *pulled* when action 1 is applied; $P : \mathbb{S} \times \mathbb{A} \times \mathbb{S} \to [0, 1]$ is the transition kernel, where $P(s, a, s')$ is the probability of transitioning from state $s$ to state $s'$ upon taking action $a$; $r = \{r(s, a)\}_{s \in \mathbb{S}, a \in \mathbb{A}}$ is the reward function, where $r(s, a)$ is the reward for taking action $a$ in state $s$. Throughout the paper, we assume that the transition kernel $P$ is unichain [Put05]; that is, under any Markov policy for this single-armed MDP $(\mathbb{S}, \mathbb{A}, P, r)$, the induced Markov chain has a single recurrent class. The unichain assumption is standard in most existing work on restless bandits [WW90, GGY20, GGY22, Ver16]. We will discuss relaxing the unichain assumption in Appendix D.

In the above setting, we are subject to a *budget constraint* that exactly $\alpha N$ arms must be activated in each time step, where $\alpha \in (0, 1)$ is a given constant and $\alpha N$ is assumed to be an integer for simplicity. This $N$-armed RB problem can be represented by the tuple $(N, \mathbb{S}^N, \mathbb{A}^N, P, r, \alpha N)$.

A policy $\pi$ for the $N$-armed problem specifies the action for each of the $N$ arms based on the history of states and actions. Under policy $\pi$, let $S_i^\pi(t) \in \mathbb{S}$ denote the state of the $i$th arm at time $t$, and we

call $\boldsymbol{S}^\pi(t) \triangleq (S_i^\pi(t))_{i\in[N]} \in \mathbb{S}^N$ the *system state*. Similarly, let $A_i^\pi(t) \in \mathbb{A}$ denote the action applied to the $i$th arm at time $t$, and let $\boldsymbol{A}^\pi(t) \triangleq (A_i^\pi(t))_{i\in[N]} \in \mathbb{A}^N$ denote the joint action vector.

The controller's goal is to find a policy that maximizes the long-run average of the total expected reward from all $N$ arms, subject to the budget constraint, assuming full knowledge of the model:

$$\underset{\text{policy } \pi}{\text{maximize}} \quad V_N^\pi \triangleq \lim_{T\to\infty} \frac{1}{T} \sum_{t=0}^{T-1} \frac{1}{N} \sum_{i=1}^N \mathbb{E}\left[r(S_i^\pi(t), A_i^\pi(t))\right] \tag{1}$$

$$\text{subject to} \quad \sum_{i=1}^N A_i^\pi(t) = \alpha N, \quad \forall t \geq 0. \tag{2}$$

Under the unichain assumption, the value $V_N^\pi$ of any policy $\pi$ is independent of the initial state. Let $V_N^* \triangleq \sup_\pi V_N^\pi$ denote the optimal value. The optimality gap of $\pi$ is defined as $V_N^* - V_N^\pi$. We say a policy $\pi$ is *asymptotically optimal* if its optimality gap converges to 0 as $N \to \infty$.

Classical theory guarantees that for a finite-state Markov decision process like an RB, there exists an optimal policy that is Markovian and stationary [Put05]. Nevertheless, the policies we propose are not Markovian policies; rather, they have internal states. Under such a policy $\pi$, the system state $\boldsymbol{S}^\pi(t)$ together with the internal state form a Markov chain, and the action $\boldsymbol{A}^\pi(t)$ depends on both the system and internal states. We design a policy such that this Markov chain has a stationary distribution. Let $\boldsymbol{S}^\pi(\infty)$ and $\boldsymbol{A}^\pi(\infty)$ denote the random elements that follow the stationary distributions of $\boldsymbol{S}^\pi(t)$ and $\boldsymbol{A}^\pi(t)$, respectively. Then the average reward of $\pi$ is equal to $V_N^\pi = \frac{1}{N} \sum_{i=1}^N \mathbb{E}\left[r(S_i^\pi(\infty), A_i^\pi(\infty))\right]$.

In later sections, when the context is clear, we drop the superscript $\pi$ from $S_i^\pi$ and $A_i^\pi$.

## 3 `Follow-the-Virtual-Advice`: A simulation-based framework

In this section, we present our framework, `Follow-the-Virtual-Advice` (FTVA). We first describe a *single-armed problem*, which involves an "average arm" from the original $N$-armed problem. We then use the optimal single-armed policy to construct the proposed policy $\text{FTVA}(\bar{\pi}^*)$.

### 3.1 Single-armed problem

The single-armed problem involves the MDP $(\mathbb{S}, \mathbb{A}, P, r)$ associated with a single arm (say arm 1 without loss of generality), where the budget is satisfied *on average*. Formally, consider the problem:

$$\underset{\text{policy } \bar{\pi}}{\text{maximize}} \quad V_1^{\bar{\pi}} \triangleq \lim_{T\to\infty} \frac{1}{T} \sum_{t=0}^{T-1} \mathbb{E}\left[r(S_1^{\bar{\pi}}(t), A_1^{\bar{\pi}}(t))\right] \tag{3}$$

$$\text{subject to} \quad \lim_{T\to\infty} \frac{1}{T} \sum_{t=0}^{T-1} \mathbb{E}\left[A_1^{\bar{\pi}}(t)\right] = \alpha. \tag{4}$$

The constraint (4) stipulates that the *average* rate of applying the active action must equal $\alpha$. Various equivalent forms of this single-armed problem have been considered in prior work [WW90, GGY20, GGY22, Ver16].

By virtue of the unichain assumption, the single-armed problem can be equivalently rewritten as the following linear program, where each decision variable $y(s,a)$ represents the steady-state probability that the arm is in state $s$ taking action $a$:

$$\underset{\{y(s,a)\}_{s\in\mathbb{S}, a\in\mathbb{A}}}{\text{maximize}} \quad \sum_{s\in\mathbb{S}, a\in\mathbb{A}} r(s,a)y(s,a) \tag{LP}$$

$$\text{subject to} \quad \sum_{s\in\mathbb{S}} y(s,1) = \alpha \tag{5}$$

$$\sum_{s'\in\mathbb{S}, a\in\mathbb{A}} y(s',a)P(s',a,s) = \sum_{a\in\mathbb{A}} y(s,a), \ \forall s \in \mathbb{S} \tag{6}$$

$$\sum_{s\in\mathbb{S}, a\in\mathbb{A}} y(s,a) = 1, \quad y(s,a) \geq 0, \ \forall s \in \mathbb{S}, a \in \mathbb{A}. \tag{7}$$

---

**Algorithm 1** `Follow-the-Virtual-Advice` (FTVA): A simulation-based framework

---

**Input:** $N$-armed problem $(N, \mathbb{S}^N, \mathbb{A}^N, P, r, \alpha N)$, initial states $\boldsymbol{S}(0)$, single-armed policy $\bar{\pi}$

**Initialize:** Virtual states $\widehat{\boldsymbol{S}}(0)$ are $N$ i.i.d. samples following the stationary distribution of $\bar{\pi}$

1: **for** $t = 0, 1, 2, \ldots$ **do**
2:      Independently sample $\widehat{A}_i(t) \leftarrow \bar{\pi}(\cdot | \widehat{S}_i(t))$ for each arm $i \in [N]$    ▷ *Generate virtual actions*
3:      **if** $\sum_{i=1}^{N} \widehat{A}_i(t) \geq \alpha N$ **then**             ▷ *Select a set $\mathcal{A}$ of $\alpha N$ arms to activate*
4:          $\mathcal{A} \leftarrow$ a set of $\alpha N$ arms chosen from $\{i \colon \widehat{A}_i(t) = 1\}$ (any tie-breaking)
5:      **else**
6:          $\mathcal{B} \leftarrow$ a set of $\alpha N - \sum_{i=1}^{N} \widehat{A}_i(t)$ arms chosen from $\{i \colon \widehat{A}_i(t) = 0\}$ (any tie-breaking)
7:          $\mathcal{A} \leftarrow \{i \colon \widehat{A}_i(t) = 1\} \cup \mathcal{B}$
8:      Apply $A_i(t) = 1$ and observe $S_i(t+1)$ for each arm $i \in \mathcal{A}$
9:      Apply $A_i(t) = 0$ and observe $S_i(t+1)$ for each arm $i \notin \mathcal{A}$
10:     **for** $i = 1, 2, 3, \ldots N$ **do**                  ▷ *Progress virtual processes*
11:         **if** $\widehat{S}_i(t) = S_i(t)$ and $\widehat{A}_i(t) = A_i(t)$ **then**     ▷ *Couple virtual and real states*
12:             $\widehat{S}_i(t+1) \leftarrow S_i(t+1)$
13:         **else**
14:             Independently sample $\widehat{S}_i(t+1)$ from the distribution $P(\widehat{S}_i(t), \widehat{A}_i(t), \cdot)$

---

Here (5) corresponds to the relaxed budget constraint, (6) is the flow balance equation, and (6)–(7) guarantee that $y(s,a)$'s are valid steady-state probabilities.

By standard results for average reward MDPs [Put05], an optimal solution $\{y^*(s,a)\}_{s \in \mathbb{S}, a \in \mathbb{A}}$ to (LP) induces an optimal policy $\bar{\pi}^*$ for the single-armed problem via the following formula:

$$\bar{\pi}^*(a|s) = \begin{cases} y^*(s,a)/(y^*(s,0) + y^*(s,1)), & \text{if } y^*(s,0) + y^*(s,1) > 0, \\ 1/2, & \text{if } y^*(s,0) + y^*(s,1) = 0. \end{cases} \quad \text{for } s \in \mathbb{S}, a \in \mathbb{A}. \quad (8)$$

Let $V_1^{\mathrm{rel}} = V_1^{\bar{\pi}^*}$ be the optimal value of (LP) and the single-armed problem.

(LP) can be viewed as a relaxation of the $N$-armed problem. To see this, take any $N$-armed policy $\pi$ and set $y(s,a)$ to be the fraction of arms in state $s$ taking action $a$ in steady state under $\pi$, i.e., $y(s,a) = \mathbb{E}\big[\frac{1}{N} \sum_{i=1}^{N} \mathbb{1}_{\{S_i^{\pi}(\infty)=s, A_i^{\pi}(\infty)=a\}}\big]$. Whenever $\pi$ satisfies the budget constraint (2), $\{y(s,a)\}$ satisfies the relaxed constraint (5) and the consistency constraints (6)–(7). Therefore, the optimal value of (LP) is an upper bound of the optimal value of the $N$-armed problem: $V_1^{\mathrm{rel}} \geq V_N^*$.

### 3.2 Constructing the $N$-armed policy

We now present `Follow-the-Virtual-Advice`, a simulation-based framework for the $N$-armed problem. FTVA takes as input *any* single-armed policy $\bar{\pi}$ that satisfies the relaxed budget constraint (4) and converts it into a $N$-armed policy, denoted by FTVA($\bar{\pi}$). Of particular interest is when $\bar{\pi}$ is an optimal single-armed policy, which leads to our result on the optimality gap. Below we introduce the general framework of FTVA without imposing any restriction on the input policy $\bar{\pi}$.

The proposed policy FTVA($\bar{\pi}$) has two main components:

- *Virtual single-armed processes.* Each arm $i$ simulates a *virtual* single-armed process, whose state is denoted as $\widehat{S}_i(t)$, with action $\widehat{A}_i(t)$ chosen according to $\bar{\pi}$. To make the distinction conspicuous, we sometimes refer to the state $S_i(t)$ and action $A_i(t)$ in the original $N$-armed problem as the *real* state/action. The virtual processes associated with different arms are independent.

- *Follow the virtual actions.* At each time step $t$, we choose the real actions $A_i(t)$'s to best match the virtual actions $\widehat{A}_i(t)$'s, to the extent allowed by the budget constraint $\sum_{i=1}^{N} A_i(t) = \alpha N$.

FTVA is presented in detail in Algorithm 1. Note that we use an appropriate coupling in Algorithm 1 to ensure that the virtual processes $(\widehat{S}_i(t), \widehat{A}_i(t))$'s are independent and each follows the Markov chain induced by the single-armed policy $\bar{\pi}$. FTVA is designed to steer the real states to be close to the virtual states, thereby ensuring a small *conversion loss* $V_1^{\bar{\pi}} - V_N^{\mathrm{FTVA}(\bar{\pi})}$. Here recall that $V_1^{\bar{\pi}}$ is the

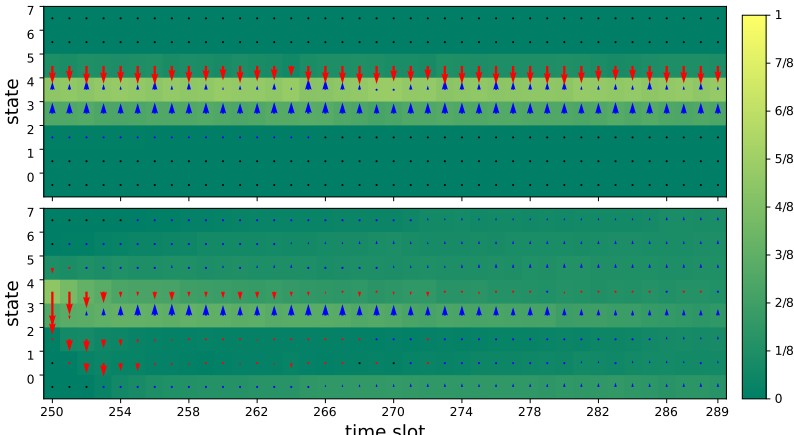

Figure 2: Time evolution of the fraction of arms in each state under LP-Priority (upper), or after switching to FTVA($\bar{\pi}^*$) (lower) since time slot 250. The x-axis represents the time slot, which ranges from 250 to 289; the y-axis represents the states; the color represents the fraction of arms in each state at each time slot. The colors and magnitudes of the arrows represent the average directions and rates at which the arms move away from each state.

average reward achieved by the input policy $\bar{\pi}$ in the single-armed problem, and that $V_N^{\texttt{FTVA}(\bar{\pi})}$ is the average reward per arm achieved by policy FTVA($\bar{\pi}$) in the $N$-armed problem.

### 3.3 Discussion on FTVA and the role of virtual processes

In this subsection, we provide insights into the mechanism of FTVA and explain the crucial role of the virtual processes. In particular, we contrast FTVA with the alternative approach of directly using the real states to choose actions, e.g., by applying the single-armed policy $\bar{\pi}^*$ to each arm's real state. We note that existing policies are essentially real-state-based, so the insights here can also explain why UGAP is necessary for existing policies to have asymptotic optimality.

We first observe that the above two approaches are equivalent in the absence of the budget constraint. In particular, even if the initial virtual state and real state of an arm $i$ are different, they will synchronize (i.e., become identical) in finite time *by chance* under mild assumptions (see Section 4.1). After this event, *if* there were no constraint, each arm $i$ will consistently follow the virtual actions, i.e., $A_i(t) = \widehat{A}_i(t) = \bar{\pi}(\widehat{S}_i(t))$ for all $t$, and the virtual states will remain identical to real states.

In the presence of the budget constraint, the arms may not remain synchronized, so the virtual processes become crucial: they guarantee that the virtual actions $\widehat{A}_i(t) = \bar{\pi}(\widehat{S}_i(t))$ nearly satisfy budget constraint, which allows the real system to approximately follow the virtual actions to remain synchronized. To see this, note that regardless of the current real states, the $N$ virtual states $\widehat{S}_1, \ldots, \widehat{S}_N$ independently follow the single-armed policy $\bar{\pi}^*$, so, in the long run, each $(\widehat{S}_i(\infty), \widehat{A}_i(\infty))$ is distributed per $y^*(\cdot, \cdot)$, the optimal solution to (LP). For large $N$, the sum $\sum_{i=1}^{N} \widehat{A}_i(\infty)$ concentrates around its expectation $N \sum_{s \in \mathbb{S}} y^*(s, 1) = \alpha N$ and thus tightly satisfy the budget constraint. In contrast, the actions generated by applying $\bar{\pi}$ to the real states are likely to significantly violate the constraint, especially when the empirical distribution of the current real states deviates from $y^*(\cdot, \cdot)$.

**An example.** We provide a concrete example illustrating the above arguments. Suppose the state space for each arm is $\mathbb{S} = \{0, 1, \ldots, 7\}$. We label action 1 as the *preferred action* for states $0, 1, 2, 3$, and action 0 for the other states. For an arm in state $s$, applying the preferred action moves the arm to state $(s + 1) \bmod 8$ with probability $p_{s,\text{R}}$, and applying the other action moves the arm to state $(s - 1)^+$ with probability $p_{s,\text{L}}$; the arm stays at state $s$ otherwise. One unit of reward is generated when the arm goes from state 7 to state 0. We assume that the budget is $N/2$ and set $\{p_{s,\text{R}}, p_{s,\text{L}}\}$ such that the optimal solution of (LP) is $y^*(s, 1) = 1/8$ for $s = 0, 1, 2, 3$ and $y^*(s, 0) = 1/8$ for $s = 4, 5, 6, 7$. That is, the optimal state distribution is uniform($\mathbb{S}$), and the optimal single-armed policy $\bar{\pi}^*$ always takes the preferred action so as to traverse state 7 as often as possible.

To see why this MDP makes the corresponding $N$-armed system tricky to control, consider the situation where $p_{s,\text{L}} >> p_{s,\text{R}}$ and most arms are in state $0$. Those arms prefer actions $1$ to move towards state $7$. However, there are only $N/2$ units of budget. If we break ties uniformly at random, each arm is not pulled with probability $1/2$ in each time slot and is likely to return to $0$ before they leave $\{0, 1, 2, 3\}$. This phenomenon can be seen from Figure 7b in Appendix G.

A similar phenomenon can be observed for LP-Priority policies. Here we consider an LP-Priority in [HF17, BS20, GGY22] that breaks ties based on the Lagrangian-optimal index. In Figure 2, we generate and visualize a sample path under LP-Priority from time 250 to 289, and contrast it with the sample path if the system switches to $\texttt{FTVA}(\bar{\pi}^*)$ from time 250 onwards. We can see that under LP-Priority, although the arms have a strong tendency to move up from state $4$ to state $5$, they move back when they reach state $5$ and thus get stuck at state $4$. In contrast, when switching to $\texttt{FTVA}(\bar{\pi}^*)$, the arrows gradually change to the correct direction, which helps the arms to escape from state $4$ and converge to the uniform distribution over the state space. Intuitively, under $\texttt{FTVA}(\bar{\pi}^*)$, an increasing number of arms

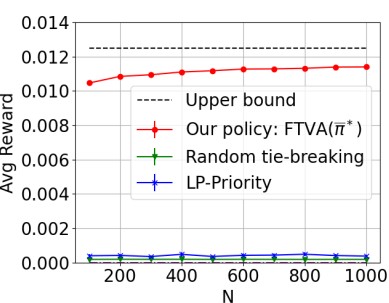

Figure 3: Comparing policies based on virtual states and real states

couple their real states with virtual states over time, which allows these arms to consistently apply the preferred actions afterward. In Appendix G.4, we include more visualizations of the policies.

In Figure 3, we compare the average reward of $\texttt{FTVA}(\bar{\pi}^*)$ with those of the random tie-breaking policy and the LP-Priority policy discussed above. We can see that $\texttt{FTVA}(\bar{\pi}^*)$ is near-optimal, while the other two policies have nearly zero rewards. Further details are provided in Appendix G. Note that while a different tie-breaking rule may solve this particular example with real states, currently there is no known rule that works in general.

## 4 Theoretical result on optimality gap

In this section, we present our main theoretical result, an upper bound on the conversion loss $V_1^{\bar{\pi}} - V_N^{\texttt{FTVA}(\bar{\pi})}$ for any given single-armed policy $\bar{\pi}$. Setting $\bar{\pi}$ to be an optimal single-armed policy $\bar{\pi}^*$ then leads to an upper bound on the optimality gap of our N-armed policy $\texttt{FTVA}(\bar{\pi}^*)$. Our result holds under the Synchronization Assumption (SA), which we formally introduce below.

### 4.1 Synchronization Assumption

SA is imposed on a given single-armed policy $\bar{\pi}$. To describe SA, we first define a two-armed system called the *leader-and-follower* system, which consists of a *leader* arm and a *follower* arm. Each arm is associated with the MDP $(\mathbb{S}, \mathbb{A}, P, r)$. At each time step $t \geq 1$, the leader arm is in state $\widehat{S}(t)$ and uses the policy $\bar{\pi}$ to chooses an action $\widehat{A}(t)$ based on $\widehat{S}(t)$; the follower arm is in state $S(t)$, and it takes the action $A(t) = \widehat{A}(t)$ regardless of $S(t)$. The state transitions of the two arms are coupled as follows. If $S(t) = \widehat{S}(t)$, then $S(t+1) = \widehat{S}(t+1)$. If $S(t) \neq \widehat{S}(t)$, then $S(t+1)$ and $\widehat{S}(t+1)$ are sampled independently from $P(S(t), A(t), \cdot)$ and $P(\widehat{S}(t), \widehat{A}(t), \cdot)$, respectively. Note that once the states of the two arms become identical, they stay identical indefinitely.

Given the initial states and actions $(S(0), A(0), \widehat{S}(0), \widehat{A}(0)) = (s, a, \widehat{s}, \widehat{a}) \in \mathbb{S} \times \mathbb{A} \times \mathbb{S} \times \mathbb{A}$, we define the *synchronization time* as the first time the two states become identical:

$$\tau^{\text{sync}}(s, a, \widehat{s}, \widehat{a}) \triangleq \min\{t \geq 0 \colon S(t) = \widehat{S}(t)\}. \tag{9}$$

**Assumption 1** (Synchronization Assumption (SA) for a policy $\bar{\pi}$)**.** We say that a single-armed policy $\bar{\pi}$ satisfies the Synchronization Assumption (SA) if for any initial states and actions $(s, a, \widehat{s}, \widehat{a}) \in \mathbb{S} \times \mathbb{A} \times \mathbb{S} \times \mathbb{A}$, the synchronization time $\tau^{\text{sync}}(s, a, \widehat{s}, \widehat{a})$ is a stopping time and satisfies

$$\mathbb{E}\left[\tau^{\text{sync}}(s, a, \widehat{s}, \widehat{a})\right] < \infty. \tag{10}$$

We view SA as an appealing alternative to UGAP for two reasons. First, there are some instances that satisfy SA but not UGAP. Two such examples are given in Figures 1 and 3. [GGY20] provides more

examples that violate UGAP, all of which can be verified to satisfy SA. Second, SA is easier to verify than UGAP. It has been acknowledged in many prior papers that UGAP lacks intuitive sufficient conditions and can only be checked numerically. In contrast, SA can be efficiently verified in several ways, which is discussed in Appendix C.

## 4.2 Bounds on conversion loss and optimality gap

We are now ready to state our main theorem.

**Theorem 1.** *Consider an $N$-armed RB problem $(N, \mathbb{S}^N, \mathbb{A}^N, P, r, \alpha N)$ under the single-armed unichain assumption. Let $\bar{\pi}$ be any single-armed policy satisfying SA. For any $N \geq 1$, the conversion loss of* `FTVA` *satisfies the upper bound*

$$V_1^{\bar{\pi}} - V_N^{\texttt{FTVA}(\bar{\pi})} \leq \frac{r_{\max} \overline{\tau}_{\max}^{\text{sync}}}{\sqrt{N}}, \tag{11}$$

*where $r_{\max} \triangleq \max_{s \in \mathbb{S}, a \in \mathbb{A}} |r(s, a)|$ and $\overline{\tau}_{\max}^{\text{sync}} \triangleq \max_{(s, a, \widehat{s}, \widehat{a}) \in \mathbb{S} \times \mathbb{A} \times \mathbb{S} \times \mathbb{A}} \mathbb{E}[\tau^{\text{sync}}(s, a, \widehat{s}, \widehat{a})]$.*

*Consequently, given any optimal single-armed policy $\bar{\pi}^*$ satisfying SA, for all any $N \geq 1$ the optimality gap of* `FTVA`$(\bar{\pi}^*)$ *is upper bounded as*

$$V_N^* - V_N^{\texttt{FTVA}(\bar{\pi}^*)} \leq V_1^{\bar{\pi}^*} - V_N^{\texttt{FTVA}(\bar{\pi}^*)} \leq \frac{r_{\max} \overline{\tau}_{\max}^{\text{sync}}}{\sqrt{N}}. \tag{12}$$

**Proof sketch.** The proof of Theorem 1 is given in Appendix E. Here we sketch the main ideas of the proof, whose key step involves bounding the conversion loss $V_1^{\bar{\pi}} - V_N^{\texttt{FTVA}(\bar{\pi})}$ using a fundamental tool from queueing theory, the Little's Law [Kle75]. Specifically, we start with the upper bound

$$V_1^{\bar{\pi}} - V_N^{\texttt{FTVA}(\bar{\pi})} = \frac{1}{N} \mathbb{E}\left[ \sum_{i=1}^N r(\widehat{S}_i(\infty), \widehat{A}_i(\infty)) - \sum_{i=1}^N r(S_i(\infty), A_i(\infty)) \right]$$

$$\leq \frac{2r_{\max}}{N} \mathbb{E}\left[ \sum_{i=1}^N \mathbb{1}\left\{ (\widehat{S}_i(\infty), \widehat{A}_i(\infty)) \neq (S_i(\infty), A_i(\infty)) \right\} \right], \tag{13}$$

which holds since the virtual process $(\widehat{S}_i(t), \widehat{A}_i(t))$ of each arm $i$ follows the single-armed policy $\bar{\pi}$. We say an arm $i$ is a *bad arm* at time $t$ if $(\widehat{S}_i(t), \widehat{A}_i(t)) \neq (S_i(t), A_i(t))$, and a *good arm* otherwise. Then $\mathbb{E}\left[ \sum_{i=1}^N \mathbb{1}\left\{ (\widehat{S}_i(\infty), \widehat{A}_i(\infty)) \neq (S_i(\infty), A_i(\infty)) \right\} \right] = \mathbb{E}[\# \text{ bad arms}]$ in steady state.

By Little's Law, we have the following relationship:

$$\mathbb{E}[\# \text{ bad arms}] = (\text{rate of generating bad arms}) \times \mathbb{E}[\text{time duration of a bad arm}].$$

It suffices to bound the two terms on the right hand side. Note that the virtual actions $\widehat{A}_i(t)$'s are i.i.d. with mean $\mathbb{E}[\widehat{A}_i(t)] = \alpha$; a standard concentration inequality shows that at most $\left| \sum_{i=1}^N \widehat{A}_i(t) - \alpha N \right| \approx O(\sqrt{N})$ bad arms are generated per time slot. On the other hand, each bad arm stays bad until its real state becomes identical to its virtual state, which occurs in $O(1)$ time by virtue of SA.

## 5 Continuous-time restless bandits

In this section, we consider the continuous-time setting. The setup, policy, and theoretical results for this setting parallel those for the discrete-time setting, except that we no longer require SA. Detailed proofs are provided in Appendix F.

**Problem setup.** The continuous-time restless bandit (CTRB) problem is similar to its discrete-time counterpart (cf. Section 2), except that each single-armed MDP runs in continuous time. In particular, an $N$-armed CTRB is given by a tuple $(N, \mathbb{S}^N, \mathbb{A}^N, G, r, \alpha N)$, where $\mathbb{S}$ is the finite state space and $\mathbb{A} = \{0, 1\}$ is the action space of a single arm. In continuous time, each arms dynamics is governed by the transition *rates* (rather than probabilities) $G = \{G(s, a, s')\}_{s, s' \in \mathbb{S}, a \in \mathbb{A}}$, where $G(s, a, s')$ is the rate of transitioning from state $s$ to state $s' \neq s$ upon taking action $a$. We again assume that the

transition kernel $G$ of each arm is unichain. Given the states and actions of all arms, the transitions of different arms are independent from each other. Similarly, $r(s, a)$ is the *instantaneous rate* of accumulating reward while taking action $a$ in state $s$. The budget constraint now requires that at any moment of time, the total number of arms taking the active action $1$ is equal to $\alpha N$.

The objective is again maximizing the long-run average reward, that is,

$$\underset{\text{policy } \pi}{\text{maximize}} \quad V_N^\pi \triangleq \lim_{T \to \infty} \frac{1}{T} \int_0^T \frac{1}{N} \sum_{i=1}^N \mathbb{E}\left[r(S_i^\pi(t), A_i^\pi(t))\right] dt \tag{14}$$

$$\text{subject to} \quad \sum_{i=1}^N A_i^\pi(t) = \alpha N \quad \forall t \geq 0. \tag{15}$$

Let $V_N^* = \sup_\pi V_N^\pi$ denote the optimal value of the above optimization problem.

**Single-armed problem.** The single-armed problem for CTRB is defined analogously as its discrete-time counterpart (3)–(4). This single-armed problem can again be written as a linear program, where the decision variable $y(s, a)$ represents the steady-state probability of being in state $s$ taking action $a$:

$$\underset{\{y(s,a)\}_{s \in \mathbb{S}, a \in \mathbb{A}}}{\text{maximize}} \quad \sum_{s \in \mathbb{S}, a \in \mathbb{A}} r(s, a) y(s, a) \tag{LP-CT}$$

$$\text{subject to} \quad \sum_{s \in \mathbb{S}} y(s, 1) = \alpha \tag{16}$$

$$\sum_{s' \in \mathbb{S}, a \in \mathbb{A}} y(s', a) G(s', a, s) = 0 \quad \forall s \in \mathbb{S} \tag{17}$$

$$\sum_{s \in \mathbb{S}, a \in \mathbb{A}} y(s, a) = 1; \quad y(s, a) \geq 0 \ \forall s \in \mathbb{S}, a \in \mathbb{A}. \tag{18}$$

In (17), we use the convention that $G(s, a, s) = -G(s, a) \triangleq -\sum_{s' \neq s} G(s, a, s')$. Again, the optimal value of (LP-CT) upper bounds the optimal value for the $N$-armed problem, i.e., $V_1^{\text{rel}} = V_1^{\bar{\pi}^*} \geq V_N^*$, where $\bar{\pi}^*$ is any optimal single-armed policy and can be computed using the same formula in (8). Note that in continuous time, the optimal policy $\bar{\pi}^*$ is carried out through uniformization [Put05].

**Our policy.** Our framework for the CTRB, `Follow-the-Virtual-Advice-CT` (FTVA-CT), is presented in Algorithm 2. `FTVA-CT` works in a similar fashion as its discrete-time counterpart. It takes a single-armed policy $\bar{\pi}$ as input, and each arm $i$ independently simulates a virtual single-armed process $(\widehat{S}_i(t), \widehat{A}_i(t))$ following $\bar{\pi}$. `FTVA-CT` then chooses the real actions $A_i(t)$'s to match the virtual actions $\widehat{A}_i(t)$'s to the extent allowed by the budget constraint $\sum_{i=1}^N A_i(t) = \alpha N$.

To run `FTVA-CT` in continuous time for the $N$-armed problem, we use the following uniformization to set discrete *decision epochs* $\{t_k\}_{k=0,1,\dots}$ for updating the actions and virtual processes. We uniformize at rate $2N g_{\max}$ with $g_{\max} = \max_{s \in \mathbb{S}, a \in \mathbb{A}} G(s, a)$, which is an upper bound on the total transition rate of the real and virtual states in an $N$-armed system. We generate a decision epoch either when a real state transitions, or when an independent exponential timer with rate $2N g_{\max} - \sum_{i=1}^N G(S_i(t), A_i(t))$ ticks.

The definitions of conversion loss, optimality gap, and asymptotic optimality are the same as the discrete-time setting.

**Conversion loss and optimality gap.** For a given single-armed policy $\bar{\pi}$, we consider a continuous-time version of the leader-and-follower system (cf. Section 4.1). For technical reasons, the initial actions are specified differently from the discrete-time setting. Specifically, we assume that the initial action $\widehat{A}(0)$ of the leader arm is chosen by $\bar{\pi}$ based on $\widehat{S}(0)$, and the follower's initial action $A(0)$ equals $\widehat{A}(0)$. As before, the follower arm always takes the same action as the leader arm regardless of its own state. Given initial states $(S(0), \widehat{S}(0)) = (s, \widehat{s})$, the synchronization time is defined as

$$\tau^{\text{sync}}(s, \widehat{s}) \triangleq \inf\{t \geq 0 \colon S(t) = \widehat{S}(t)\}. \tag{19}$$

We no longer need to impose the Synchronization Assumption, since the unichain assumption automatically implies $\mathbb{E}\left[\tau^{\text{sync}}(s, \widehat{s})\right] < \infty$ in continuous time—see Lemma 5 in Appendix F.

**Algorithm 2** `Follow-the-Virtual-Advice-CT (FTVA-CT)`

---

**Input:** $(N, \mathbb{S}^N, \mathbb{A}^N, G, r, \alpha N)$, initial states $\boldsymbol{S}(0)$, single-armed policy $\bar{\pi}$, max transition rate $g_{\max}$
**Initialize:** Virtual states $\widehat{\boldsymbol{S}}(0)$ are $N$ i.i.d. samples following the stationary distribution of $\bar{\pi}$; $t_0 = 0$

1:  **for** $k = 0, 1, 2, \ldots$ **do**
2:      Independently sample $\widehat{A}_i(t_k) \leftarrow \bar{\pi}(\cdot|\widehat{S}_i(t_k))$ for each arm $i \in [N]$ ▷ *Generate virtual actions*
3:      **if** $\sum_{i=1}^N \widehat{A}_i(t_k) \geq \alpha N$ **then**                           ▷ *Select a set $\mathcal{A}$ of $\alpha N$ arms to activate*
4:          $\mathcal{A} \leftarrow$ a set of $\alpha N$ arms chosen from $\{i : \widehat{A}_i(t_k) = 1\}$ (any tie-breaking)
5:      **else**
6:          $\mathcal{B} \leftarrow$ a set of $\alpha N - \sum_{i=1}^N \widehat{A}_i(t_k)$ arms chosen from $\{i : \widehat{A}_i(t_k) = 0\}$ (any tie-breaking)
7:          $\mathcal{A} \leftarrow \{i : \widehat{A}_i(t_k) = 1\} \cup \mathcal{B}$
8:      Apply $A_i(t_k) = 1$ for each arm $i \in \mathcal{A}$, and apply $A_i(t_k) = 0$ for each arm $i \notin \mathcal{A}$
9:      ▷ *The section below progresses the system to the next decision epoch and updates states*
10:      $g_k^{\text{real}} \leftarrow \sum_{i=1}^N G(S_i(t_k), A_i(t_k))$
11:      $\widetilde{t}_{k+1} \leftarrow t_k +$ a sample from the distribution $\text{Exp}(2Ng_{\max} - g_k^{\text{real}})$
12:      **if** there is an arm $i^*$ whose real state transitions before $\widetilde{t}_{k+1}$ **then**
13:          $(t_{k+1}, S_{i^*}(t_{k+1})) \leftarrow$ (transition time, new real state)
14:          **if** $\widehat{S}_{i^*}(t_k) = S_{i^*}(t_k)$ and $\widehat{A}_{i^*}(t_k) = A_{i^*}(t_k)$ **then**        ▷ *Couple virtual and real states*
15:              $\widehat{S}_{i^*}(t_{k+1}) \leftarrow S_{i^*}(t_{k+1})$
16:      **else**                          ▷ *No transitions or transitions of virtual states of uncoupled arms*
17:          $t_{k+1} \leftarrow \widetilde{t}_{k+1}$
18:          $g_k^{\text{virtual}} \leftarrow \sum_{i=1}^N G(\widehat{S}_i(t_k), \widehat{A}_i(t_k)) \mathbb{1}\left\{\widehat{S}_i(t_k) \neq S_i(t_k) \text{ or } \widehat{A}_i(t_k) \neq A_i(t_k)\right\}$
19:          With probability $1 - \frac{g_k^{\text{virtual}}}{2Ng_{\max} - g_k^{\text{real}}}$, **continue**
20:          Sample a pair $(i^*, s') \in [N] \times \mathbb{S}$ with probability $\frac{G(\widehat{S}_{i^*}(t_k), \widehat{A}_{i^*}(t_k), s') \cdot \mathbb{1}\left\{\widehat{S}_{i^*}(t_k) \neq s'\right\}}{2Ng_{\max} - g_k^{\text{real}}}$
21:          **if** $\widehat{S}_{i^*}(t_k) \neq S_{i^*}(t_k)$ or $\widehat{A}_{i^*}(t_k) \neq A_{i^*}(t_k)$ **then** $\widehat{S}_{i^*}(t_{k+1}) \leftarrow s'$
22:      Other real and virtual states stay unchanged

---

**Theorem 2.** *Consider an $N$-armed CTRB $(N, \mathbb{S}^N, \mathbb{A}^N, G, r, \alpha N)$ under the single-armed unichain assumption. For any single-armed policy $\bar{\pi}$, the conversion loss of* `FTVA-CT` *is upper bounded as*

$$V_1^{\bar{\pi}} - V_N^{\text{FTVA-CT}(\bar{\pi})} \leq \frac{r_{\max}(1 + 2g_{\max}\overline{\tau}_{\max}^{\text{sync}})}{\sqrt{N}}, \quad \forall N \geq 1, \tag{20}$$

*where $r_{\max} = \max_{s \in \mathbb{S}, a \in \mathbb{A}} |r(s,a)|$, $\overline{\tau}_{\max}^{\text{sync}} = \max_{s \in \mathbb{S}, \widehat{s} \in \mathbb{S}} \mathbb{E}\left[\tau^{\text{sync}}(s, \widehat{s})\right]$.*

*Consequently, for any optimal single-armed policy $\bar{\pi}^*$, the optimality gap of* `FTVA-CT`$(\bar{\pi}^*)$ *satisfies*

$$V_N^* - V_N^{\text{FTVA-CT}(\bar{\pi}^*)} \leq V^{\bar{\pi}^*} - V_N^{\text{FTVA-CT}(\bar{\pi}^*)} \leq \frac{r_{\max}(1 + 2g_{\max}\overline{\tau}_{\max}^{\text{sync}})}{\sqrt{N}}, \quad \forall N \geq 1. \tag{21}$$

Theorem 2 establishes an $O(1/\sqrt{N})$ optimality gap without requiring UGAP or any additional assumptions beyond the standard unichain condition.

## 6  Conclusion

In this paper, we study the average-reward restless bandit problem. We propose a simulation-based framework called `Follow-the-Virtual-Advice` that converts a single-armed optimal policy into a policy in the original $N$-arm system with an $O(1/\sqrt{N})$ optimality gap. In the discrete-time setting, our results hold under the Synchronization Assumption (SA), a mild and easy-to-verify assumption that covers some problem instances that do not satisfy UGAP. In the continuous-time setting, our results do not require any additional assumptions beyond the standard unichain condition. In both settings, our work is the first to achieve asymptotic optimality without assuming UGAP. It will be interesting in future work to explore the possibility of achieving optimality gaps smaller than $O(1/\sqrt{N})$ without relying on UGAP or the non-degenerate condition.

**Acknowledgement.** Y. Hong and W. Wang are supported in part by NSF grants CNS-200773 and ECCS-2145713. Q. Xie is supported in part by NSF grant CNS-1955997 and a J.P. Morgan Faculty Research Award. Y. Chen is supported in part by NSF grants CCF-1704828 and CCF-2233152. This project started when Q. Xie, Y. Chen, and W. Wang were attending the Data-Driven Decision Processes program of the Simons Institute for the Theory of Computing.

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

# Appendices

## A  More related work

In this section, we provide additional discussion on related work.

**Relationship between the infinite-horizon average-reward setting and the finite-horizon to­tal-reward/infinite-horizon discounted-reward setting.**  While infinite-horizon average-reward RBs, finite-horizon total-reward RBs, and infinite-horizon discounted-reward RBs are three different settings, one may think that the approaches in the other two settings apply to the infinite-horizon average-reward setting by taking a large enough horizon in the finite-horizon setting or letting the discount factor to go to 1 in the infinite-horizon discounted-reward setting. Here we briefly discuss the fundamental difference between the infinite-horizon average-reward setting and the other two settings for RB problems. For simplicity, we will refer to the other two settings as the finite-horizon/discounted-reward setting.

From the algorithm design perspective, the finite-horizon/discounted-reward setting focuses on optimizing the transient performance, while the infinite-horizon average reward setting focuses on optimizing the steady-state performance. This distinction has a big impact on the algorithm design and the complexity. To see this, we note that both the finite-horizon/discounted-reward setting [HF17, ZCJW19, BS20, GNJT23, ZF21, ZF22, GGY22] and the infinite-horizon average-reward setting [Whi88, WW90, Ver16, GGY20, GGY22] utilize certain forms of linear programming (LP) relaxations to design algorithms. However, the LPs utilized differ substantially. In the finite-horizon/discounted-reward setting, the LP optimizes the future trajectory for a predetermined number of time steps. In contrast, in the infinite-horizon average-reward setting, the LP solves for the optimal state-action frequency in steady state, which can be seen as a fixed point rather than a trajectory. As a result, the number of variables and constraints of the LP for the finite-horizon/discounted-reward setting scales with the number of time steps, which is not the case for the infinite-horizon average-reward setting. On the other hand, because the LP relaxation for the infinite-horizon average-reward setting does not take into account the transient behavior, it provides less information, making the algorithm design more tricky. Consequently, the policies in prior work require the strong assumption of UGAP to achieve asymptotic optimality.

From the analysis perspective, the infinite-horizon average-reward setting requires a more careful analysis of the long-term effect of actions. To see this, note that the optimality gap bounds in the finite-horizon/discounted-reward setting have at least a *quadratic* dependency on the (effective) horizon [see, e.g. ZF22, GGY22]. Consequently, when translated to the bounds on the optimality gap of average reward per time slot, those bounds diverge to infinity as the time horizon goes to infinity.

**Relationship between the RB problem and other bandit problems.**  While the exact optimality of the RB problem is in general intractable, there is a special case that has been solved optimally. Specifically, consider the case when an arm stays in the same state when it is not pulled, and only one arm is pulled at a time. The optimal policy for this special case is the celebrated Gittins index policy [GJ74, Git79]. A more recent reference on this topic is [GGW11].

The RB problem falls within the broader class of bandit problems, for which different formulations exist including stochastic bandits, adversarial bandits, and Bayesian bandits. The common theme in these formulations is to find a reward-maximizing strategy of pulling arms in the presence of uncertainty in the arms' rewards; see the book [LS20] for a comprehensive overview. Among these formulations, closely related to RBs is the Bayesian bandit problem, where Bayesian posteriors are used to model knowledge of unknown reward distributions. The Bayesian posterior can be seen as a state with known transition probabilities, hence the Bayesian bandit problem can be analyzed by applying tools from RBs. Examples demonstrating this connection can be found in [BS20].

## B  Discussion on the computational costs of `FTVA`

`FTVA` defined in Algorithm 1 can be computed and implemented efficiently.

First, computing `FTVA` requires solving (LP) once. Given that (LP) does not depend on $N$, the computational cost here is a constant.

Next, we examine the computational cost of implementing `FTVA`.

- Each arm simulates a virtual process. Because the computation required for each simulation does not scale with $N$, the overall computational cost here is linear in $N$.
- In each time step, `FTVA` selects arms from a subset of arms to determine the real actions. The computational cost for this selection process scales linearly with $N$.

Combining these components, we can see that the computational cost of implementing `FTVA` is linear in $N$.

Additionally, we note that the simulation of the virtual processes is independent across the arms, so they can be implemented in a fully distributed manner. However, when taking real actions, the policy needs to know the virtual actions from all arms, so this step cannot be implemented distributedly.

## C   Sufficient conditions for Synchronization Assumption (SA)

Recall that for the discrete-time setting, our bound on the conversion loss in Theorem 1 holds when the input single-armed policy $\bar{\pi}$ satisfies the Synchronization Assumption (SA). SA stipulates that the two arms in the leader-and-follower system (cf. Section 4.1) under the policy $\bar{\pi}$ will reach the same state in finite expected time. All other existing work requires the UGAP assumption, which pertains to the global behavior of a non-linear dynamic system and has no known sufficient conditions that are easily checkable. In comparison, SA is a reachability assumption imposed on a *finite state Markov chain* and thus substantially simpler.

To provide a more intuitive understanding of SA, in this section we present several sufficient conditions for SA to hold. These conditions involve the recurrent classes in the Markov chains induced by certain single-armed policies as well as the existence of self-loops or cycles in the corresponding transition diagrams. Such conditions can be verified, often in a straightforward manner, by inspecting the transition probabilities. As self-loops and cycles are common in many classes of MDPs, these conditions showcase that SA is a relatively mild assumption. We emphasize that the sufficient conditions presented in this section are not exhaustive. They serve as illustrative examples of when SA holds and offer insights on the nature of synchronization.

The discussion in this section pertains to the discrete-time setting. We reiterate that our results for the continuous-time setting do not require SA.

### C.1   Preliminaries

Before presenting the sufficient conditions, we first prove a few preliminary facts. The first one is a basic property of unichain MDPs.

**Proposition 1.** *Consider two arbitrary Markovian policies $\bar{\pi}$ and $\bar{\pi}'$ for the unichain MDP $(\mathbb{S}, \mathbb{A}, P, r)$. Let the recurrent classes of the two policies $\bar{\pi}$ and $\bar{\pi}'$ be $\mathcal{S}$ and $\mathcal{S}'$, respectively. Then $\mathcal{S} \cap \mathcal{S}' \neq \emptyset$.*

*Proof.* We prove this proposition by contradiction. Suppose $\mathcal{S} \cap \mathcal{S}' = \emptyset$. We define a new policy $\bar{\pi}''$ as

$$\bar{\pi}''(a|s) = \begin{cases} \bar{\pi}(a|s), & \text{if } s \in \mathcal{S}, \\ \bar{\pi}'(a|s), & \text{otherwise.} \end{cases} \quad \text{for } s \in \mathbb{S}, a \in \mathbb{A}.$$

Then $\mathcal{S}$ and $\mathcal{S}'$ are two distinct recurrent classes under $\bar{\pi}''$, because by definition of $\bar{\pi}''$, an arm with the initial state in $\mathcal{S}$ remains in $\mathcal{S}$, so it cannot reach $\mathcal{S}'$; similarly an arm cannot reach $\mathcal{S}$ from $\mathcal{S}'$. The existence of two recurrent classes under $\bar{\pi}''$ contradicts the unichain condition. Therefore, we must have $\mathcal{S} \cap \mathcal{S}' \neq \emptyset$. □

The next proposition provides a convenient way for verifying SA and is used for establishing other sufficient conditions.

**Proposition 2.** *Consider the leader-and-follower system under the policy $\bar{\pi}$. If there exists some $t < \infty$ such that for all initial states and actions $(s, a, \widehat{s}, \widehat{a}) \in \mathbb{S} \times \mathbb{A} \times \mathbb{S} \times \mathbb{A}$,*

$$\mathbb{P}\left(\tau^{\text{sync}}(s, a, \widehat{s}, \widehat{a}) < t\right) > 0,$$

*then Synchronization Assumption holds for the policy $\bar{\pi}$.*

*Proof.* We claim that for any $(s, a, \widehat{s}, \widehat{a}) \in \mathbb{S} \times \mathbb{A} \times \mathbb{S} \times \mathbb{A}$,

$$\mathbb{P}\left(\tau^{\text{sync}}(s, a, \widehat{s}, \widehat{a}) \geq kt\right) \leq \left(\max_{s', a', \widehat{s}', \widehat{a}'} \mathbb{P}\left(\tau^{\text{sync}}(s', a', \widehat{s}', \widehat{a}') \geq t\right)\right)^k, \tag{22}$$

where the $\max_{s', a', \widehat{s}', \widehat{a}'}$ is taken over $\mathbb{S} \times \mathbb{A} \times \mathbb{S} \times \mathbb{A}$. We prove this claim by induciton on $k$. The base case of $k = 1$ is given. Suppose we have proved (22) for a certain $k$, then

$$\begin{aligned}
&\mathbb{P}\left(\tau^{\text{sync}}(s, a, \widehat{s}, \widehat{a}) \geq (k+1)t\right) \\
&= \mathbb{P}\left(\tau^{\text{sync}}(s, a, \widehat{s}, \widehat{a}) \geq kt\right) \cdot \mathbb{P}\left(\tau^{\text{sync}}(s, a, \widehat{s}, \widehat{a}) \geq (k+1)t \mid \tau^{\text{sync}}(s, a, \widehat{s}, \widehat{a}) \geq kt\right) \\
&= \mathbb{P}\left(\tau^{\text{sync}}(s, a, \widehat{s}, \widehat{a}) \geq kt\right) \cdot \mathbb{P}\left(\tau^{\text{sync}}(S(kt), A(kt), \widehat{S}(kt), \widehat{A}(kt)) \geq t \mid S(kt) \neq \widehat{S}(kt)\right) \\
&\leq \mathbb{P}\left(\tau^{\text{sync}}(s, a, \widehat{s}, \widehat{a}) \geq kt\right) \cdot \max_{s', a', \widehat{s}', \widehat{a}'} \mathbb{P}\left(\tau^{\text{sync}}(s', a,' \widehat{s}', \widehat{a}') \geq t\right) \\
&\leq \left(\max_{s', a', \widehat{s}', \widehat{a}'} \mathbb{P}\left(\tau^{\text{sync}}(s', a', \widehat{s}', \widehat{a}') \geq t\right)\right)^{k+1}.
\end{aligned}$$

where in the second equality we have used the Markov property of the system, and in the last inequality we apply the induction hypothesis. We have proved (22).

The bound on the expectation $\mathbb{E}\left[\tau^{\text{sync}}(s, a, \widehat{s}, \widehat{a})\right]$ follows by summing the tail bound (22) over $k$:

$$\begin{aligned}
\mathbb{E}\left[\tau^{\text{sync}}(s, a, \widehat{s}, \widehat{a})\right] &\leq \sum_{k=0}^{\infty} t \mathbb{P}\left(\tau^{\text{sync}}(s, a, \widehat{s}, \widehat{a}) \geq kt\right) \\
&\leq \frac{t}{1 - \max_{s', a', \widehat{s}', \widehat{a}'} \mathbb{P}\left(\tau^{\text{sync}}(s', a', \widehat{s}', \widehat{a}') \geq t\right)} \\
&= \frac{t}{\min_{s', a', \widehat{s}', \widehat{a}'} \mathbb{P}\left(\tau^{\text{sync}}(s', a', \widehat{s}', \widehat{a}') < t\right)} < \infty.
\end{aligned}$$

We have established SA and thereby finished the proof. $\qquad \square$

*Remark* 1. Proposition 2 implies that SA only requires that the Markov chain of the leader-and-follower system, $(S(t), \widehat{S}(t))$, can reach the subset of states $\{(s, \widehat{s}) \in \mathbb{S}^2 \colon s = \widehat{s}\}$ from any initial state. As a result, SA can be efficiently verified using path-finding graph algorithms on the state transition diagram of the Markov chain.

## C.2 Sufficient conditions based on self-loops

The unichain condition and Proposition 1 guarantee that there are some states that the leader arm and the follower arm will both visit for a positive fraction of times.[2] If we can further show that the two arms have a positive probability to visit one of those states *at the same time*, then Proposition 2 can be applied to verify SA. A natural sufficient condition is the existence of self-loops, which guarantees that with a positive probability, the arm reaching those states first will wait for the other arm to come to the same state.

**Proposition 3.** *Consider the single-armed problem with the unichain MDP $(\mathbb{S}, \mathbb{A}, P, r)$ and budget $\alpha \in (0, 1)$. Let $\bar{\pi}$ be a single-armed policy with recurrent class $\mathcal{S}$. If $P(s, 0, s) > 0$ and $P(s, 1, s) > 0$ for all $s \in \mathcal{S}$, then Synchronization Assumption holds for the policy $\bar{\pi}$.*

Proposition 3 assumes that all states in the recurrent class of the policy $\bar{\pi}$ have self-loops. We can actually require fewer self-loops if we can characterize the recurrent states of the follower arm. In particular, let $\bar{\pi}_1$ denote the *all-one policy*, the policy that applies action 1 in all states. Observe that if the leader arm applies action 1 for a sufficiently long time, the follower arm will also apply action 1 and effectively follows $\bar{\pi}_1$. The recurrent class of the policy $\bar{\pi}_1$ must intersect the recurrent class of $\bar{\pi}$, so it suffices to have self-loops in the intersection of these two recurrent classes. The idea is formalized in Proposition 4.

---

[2]Although the follower arm does not follow a Markovian policy since it copies actions from the leader arm, by standard results on average reward MDP, the set of states in which the follower arm spends a positive fraction of time is the same as the recurrent class of some Markovian policy (cf. Chapter 8.9 of [Put05]). This class of states must intersect with $\mathcal{S}$ by Proposition 1 and the unichain condition.

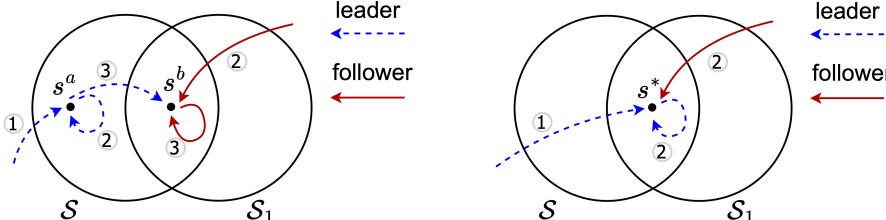

Figure 4: The illustration of the positive probability sample paths that lead to synchronization in Proposition 4 (left) and Proposition 5 (right). In each figure, the dotted arrows correspond to the sample path of the leader arm's state, while the solid arrows correspond to the sample path of the follower arm's state. The numbers near the arrows denote the temporal order of the transition events.

**Proposition 4.** *Consider the single-armed problem with the unichain MDP $(\mathbb{S}, \mathbb{A}, P, r)$ and budget $\alpha \in (0,1)$. Let $\bar{\pi}$ be a single-armed policy with recurrent class $\mathcal{S}$. Let the recurrent class of the all-one policy $\bar{\pi}_1$ be denoted as $\mathcal{S}_1$. If the following conditions hold:*

  1. *There exists $s^a \in \mathcal{S}$ such that $\bar{\pi}(1|s^a) > 0$ and $P(s^a, 1, s^a) > 0$;*

  2. *There exists $s^b \in \mathcal{S} \cap \mathcal{S}_1$ such that $P(s^b, 0, s^b) > 0$ and $P(s^b, 1, s^b) > 0$,*

*then Synchronization Assumption holds for the policy $\bar{\pi}$.*

Proposition 5 below gives another sufficient condition, which only requires one self-loop. Note that this condition does not subsume the one in Proposition 4.

**Proposition 5.** *Consider the single-armed problem with the unichain MDP $(\mathbb{S}, \mathbb{A}, P, r)$ and budget $\alpha \in (0,1)$. Let $\bar{\pi}$ be a single-armed policy with recurrent class $\mathcal{S}$. Let the recurrent class of the all-one policy $\bar{\pi}_1$ be denoted as $\mathcal{S}_1$. If there exists $s^* \in \mathcal{S} \cap \mathcal{S}_1$ such that $\bar{\pi}(1|s^*) > 0$ and $P(s^*, 1, s^*) > 0$, then Synchronization Assumption holds for the policy $\bar{\pi}$.*

We remark that Propositions 4 and 5 have analogous versions that are stated in terms of the all-zero policy $\bar{\pi}_0$, the policy that applies action 0 in all states. We omit the details.

Now we prove the above propositions. Note that Proposition 3 is strictly weaker than Proposition 4: because the single-armed policy $\bar{\pi}$ satisfies the budget constraint with $\alpha \in (0,1)$, there must be a state $s^a \in \mathcal{S}$ such that $\bar{\pi}(1|s^a) > 0$. Therefore, we only need to prove Propositions 4 and 5.

*Proof of Proposition 4.* Given any initial states and actions $(S(0), A(0), \widehat{S}(0), \widehat{A}(0)) = (s, a, \widehat{s}, \widehat{a}) \in \mathbb{S} \times \mathbb{A} \times \mathbb{S} \times \mathbb{A}$, we construct a sequence of positive probability events that leads to $S(t) = \widehat{S}(t)$:

  1. The leader arm reaches the state $s^a \in \mathcal{S}$ after $t_1$ time slots;

  2. The leader arm stays in state $s^a$ and keeps applying action 1 for $t_2$ times slots; meanwhile, the follower arm also applies action 1 and reaches the state $s^b \in \mathcal{S} \cap \mathcal{S}_1$;

  3. The leader arm reaches the state $s^b$ in another $t_3$ time slots; meanwhile, the follower arm stays in the state $s^b$, so the two arms synchronize.

The transitions of the two arms during the above sequence of events are illustrated in Figure 4. We argue that there exists suitably large $t_1, t_2, t_3$ such that the above three events happen with a positive probability. The first event can happen for a suitably large $t_1$ because the leader arm follows the policy $\bar{\pi}$, and $s^a$ is in the recurrent class of $\bar{\pi}$. In the second event, the leader arm can stay in state $s^a$ and keep applying action 1 because $\bar{\pi}(1|s^a) > 0$ and $P(s^a, 1, s^a) > 0$. The follower arm can reach $s^b$ after a suitably large $t_2$ time slots because it keeps applying action 1 and $s^b$ is in the recurrent class of the all-one policy $\bar{\pi}_1$. In the third event, the leader arm can reach $s^b$ after a suitably large $t_3$ time slots because $s^b$ is also in the recurrent class of $\bar{\pi}$. Meanwhile, the follower arm can stay in $s^b$ because $P(s^b, 0, s^b) > 0$ and $P(s^b, 1, s^b) > 0$. Therefore, we have proved that for any $(s, a, \widehat{s}, \widehat{a})$,

$$\mathbb{P}\left(\tau^{\mathrm{sync}}(s, a, \widehat{s}, \widehat{a}) \leq t_1 + t_2 + t_3\right) > 0.$$

By Proposition 2, we establish SA. □

*Proof of Proposition 5.* Given any initial states and actions $(S(0), A(0), \widehat{S}(0), \widehat{A}(0)) = (s, a, \widehat{s}, \widehat{a}) \in \mathbb{S} \times \mathbb{A} \times \mathbb{S} \times \mathbb{A}$, we construct a sequence of positive probability events that leads to $S(t) = \widehat{S}(t)$:

1. The leader arm reaches the state $s^*$ after $t_1$ time slots;

2. The leader arm stays in the state $s^*$ and keeps applying action 1 for $t_2$ time slots; meanwhile, the follower arm also applies action 1 and reaches the state $s^*$, so the two arms synchronize.

The transitions of the two arms during the above sequence of events are illustrated in Figure 4. We argue that there exists suitable $t_1$ and $t_2$ such that the above two events happen with a positive probability. The first event can happen for a suitably large $t_1$ because $s^*$ is in the recurrent class of $\bar{\pi}$. In the second event, the leader arm can stay in the state $s^*$ and keeps applying action 1 because $\bar{\pi}(1|s^*) > 0$ and $P(s^*, 1, s^*) > 0$. Meanwhile, the follower arm can reach $s^*$ after a suitably large $t_2$ time slots because $s^*$ is also in the recurrent class of the all-one policy $\bar{\pi}_1$. Therefore, we have proved that for any $(s, a, \widehat{s}, \widehat{a})$,

$$\mathbb{P}\left(\tau^{\mathrm{sync}}(s, a, \widehat{s}, \widehat{a}) \leq t_1 + t_2\right) > 0.$$

By Proposition 2, we establish SA. □

### C.3 Sufficient conditions based on cycles

The above sufficient conditions are based on self-loops, which can be viewed as cycles of minimal length. These conditions can be generalized to longer cycles, formally defined below.

**Definition 1** (Cycle). Consider the MDP $(\mathbb{S}, \mathbb{A}, P, r)$. We call an ordered set of states $C = (s_0, s_1, \ldots, s_L)$ a *cycle under the Markovian policy* $\bar{\pi}$ if there is a positive probability of transitioning from $s_j$ to $s_{j+1}$ for all $j$ under the policy $\bar{\pi}$ (we identify $s_{L+1}$ with $s_0$). We call the ordered set $C$ a *cycle under any policy* if there is a positive probability of transitioning from $s_j$ to $s_{j+1}$ for all $j$ under any policy. In both cases, we call $L$ the *length of the cycle $C$*.

We give two sufficient conditions based on cycles. These conditions are relaxations of the conditions in Proposition 4 and 5.

**Proposition 6.** *Consider the single-armed problem with the unichain MDP $(\mathbb{S}, \mathbb{A}, P, r)$ and budget $\alpha \in (0, 1)$. Let $\bar{\pi}$ be a single-armed policy with recurrent class $\mathcal{S}$. Let the recurrent class of the all-one policy $\bar{\pi}_1$ be denoted as $\mathcal{S}_1$. If the following conditions hold:*

1. *There exists a cycle $C^a$ under the policy $\bar{\pi}$ in $\mathcal{S}$, and $\bar{\pi}(1|s) > 0$ for all $s \in C^a$;*

2. *There exists cycle $C^b$ under any policy in $\mathcal{S} \cap \mathcal{S}_1$;*

3. *The lengths of the two cycles $C^a$ and $C^b$ are relatively prime.*

*then Synchronization Assumption holds for the policy $\bar{\pi}$.*

**Proposition 7.** *Consider the single-armed problem with the unichain MDP $(\mathbb{S}, \mathbb{A}, P, r)$ and budget $\alpha \in (0, 1)$. Let $\bar{\pi}$ be a single-armed policy with recurrent class $\mathcal{S}$. Let the recurrent class of the all-one policy $\bar{\pi}_1$ be denoted as $\mathcal{S}_1$. If the following conditions hold:*

1. *There exists a cycle $C^*$ under the policy $\bar{\pi}$ in $\mathcal{S} \cap \mathcal{S}_1$, and $\bar{\pi}(1|s) > 0$ for all $s \in C^a$;*

2. *The policy $\bar{\pi}_1$ is aperiodic,*

*then Synchronization Assumption holds for the policy $\bar{\pi}$.*

Similarly to Propositions 4 and 5, Propositions 6 and 7 have analogous version that are stated in terms of the all-zero policy $\bar{\pi}_0$. We omit the details.

Now we prove Propositions 6 and 7.

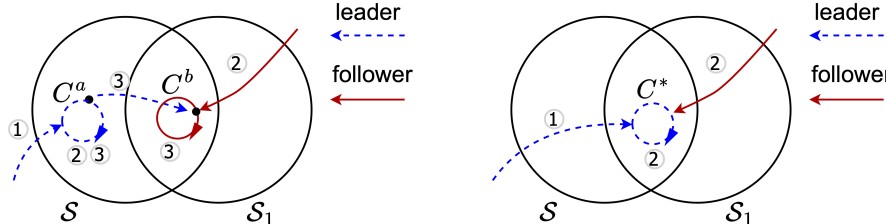

Figure 5: The illustration of the positive probability sample paths that lead to synchronization in Proposition 6 (left) and Proposition 7 (right). In each figure, the dotted arrows correspond to the sample path of the leader arm's state, while the solid arrows correspond to the sample path of the follower arm's state. The numbers near the arrows denote the temporal order of the transition events.

*Proof of Proposition 6.* Given any initial states and actions $(S(0), A(0), \widehat{S}(0), \widehat{A}(0)) = (s, a, \widehat{s}, \widehat{a}) \in \mathbb{S} \times \mathbb{A} \times \mathbb{S} \times \mathbb{A}$, we construct a sequence of positive probability events that leads to $S(t) = \widehat{S}(t)$:

1. The leader arm reaches the cycle $C^a$ after $t_1$ time slots;

2. The leader arm transitions along the cycle and keeps applying action 1; meanwhile, the follower arm also applies action 1 and reaches the cycle $C^b$; this happens in $t_2$ times slots; let the states of the leader and follower arm at this moment be $s^a$ and $s^b$;

3. The leader arm transitions along the cycle $C^a$ for another $c$ laps, for some positive integer $c$, and then spends another $t_3$ time slots to go from $s^a$ to $s^b$; meanwhile, the follower arm transitions along the cycle $C^b$ for another $d$ laps, for some positive integer $d$.

The transition of the arms during the sequence of events are as illustrated in Figure 5. We argue that there exists some suitable $t_1$, $t_2$, $t_3$, $c$, and $d$ such that the above four events happen with a positive probability, and the two arms synchronize after the four events.

The first event happens with a positive probability for suitably large $t_1$ because $C^a$ is in $\mathcal{S}$. The second event can happen for suitably large $t_2$ because $\pi(1|s) > 0$ for all $s \in C^a$, and $C^b$ is in $\mathcal{S}_1$.

After the first two events, suppose the state of the leader arm is $s^a$, and the state of the follower arm is $s^b$, then $s^a \in C^a$ and $s^b \in C^b$. Because both cycles are in $\mathcal{S}$, there exists a positive probability path from $s^a$ to $s^b$ under $\bar{\pi}$. Let $t_3$ be the length of the path. Let $L^a$ and $L^b$ be the length of cycles $C^a$ and $C^b$. Because $L^a$ and $L^b$ are relatively prime, we can take $c$ and $d$ such that $dL^b - cL^a = t_3$. With this choice of $c, d$, and $t_3$, the two arms will synchronize after the third and fourth events.

Now we argue that the third and fourth events happen with a positive probability. Because $C^a$ is a cycle under the policy $\bar{\pi}$, the leader arm can transition along the cycle for an arbitrary number of time slots. Similarly, because $C^b$ is a cycle under any policy, the follower arm can transition along the cycle for an arbitrary number of time slots; the leader arm can reach $s^b$ from $s^a$ because of the positive probability path described in the last paragraph.

Therefore, we have proved that for any $(s, a, \widehat{s}, \widehat{a})$,

$$\mathbb{P}\left(\tau^{\mathrm{sync}}(s, a, \widehat{s}, \widehat{a}) \leq t_1 + t_2 + cL^a + t_3\right) > 0.$$

By Proposition 2, we establish SA. $\square$

*Proof of Proposition 7.* Given any initial states and actions $(S(0), A(0), \widehat{S}(0), \widehat{A}(0)) = (s, a, \widehat{s}, \widehat{a}) \in \mathbb{S} \times \mathbb{A} \times \mathbb{S} \times \mathbb{A}$, we construct a sequence of positive probability events that leads to $S(t) = \widehat{S}(t)$:

1. The leader arm reaches the cycle $C^*$ after $t_1$ time slots;

2. The leader arm transitions along the cycle and keeps applying action 1; meanwhile, the follower arm also applies action 1 and reaches the same state as the leader arm after $t_2$ time slots.

The transition of the arms during the sequence of events are as illustrated in Figure 5. We argue that the above two events happen with a positive probability for suitable $t_1$ and $t_2$. The first event can happen for suitably large $t_1$ because the leader arm follows the policy $\bar{\pi}$ the cycle $C^*$ is in the recurrent class of $\bar{\pi}$. After reaching $C^*$, the leader arm can remain on $C^*$ because $C^*$ is a cycle under the policy $\bar{\pi}$, and $\bar{\pi}(1|s) > 0$ for all $s \in C^*$. The follower arm can reach the same state as the leader arm because the cycle $C^*$ is also in the recurrent class of $\bar{\pi}_1$, and $\bar{\pi}_1$ is aperiodic. Therefore, we have proved that for any $(s, a, \hat{s}, \hat{a})$,

$$\mathbb{P}\left(\tau^{\text{sync}}(s, a, \hat{s}, \hat{a}) \leq t_1 + t_2\right) > 0.$$

By Proposition 2, we establish SA. $\qquad\square$

## D   On relaxing the unichain condition

Throughout this paper, we have imposed the (all-policy) unichain condition, i.e., all Markovian single-armed policies have a single recurrent class. Making this blanket assumption simplifies our presentation, but it can be substantially relaxed for many of our results, as we discuss below.

The main change caused by dropping the unichain assumption is the definition of the single-armed problem, because there could be multiple recurrent classes under a single-armed policy, each corresponding to a different long-run average reward. To adapt to this change, we can let both the single-armed policy $\bar{\pi}$ and the initial state distribution $\mu$ be optimization variables of the single-armed problem. We denote the average reward given $\bar{\pi}$ and $\mu$ as $V_1^{\bar{\pi},\mu}$. We still denote the optimal value of the single-armed problem as $V_1^{\text{rel}}$. Note that the single-armed problem can still be solved by the same linear program in (LP) or (LP-CT): The optimal initial state distribution $\mu^*$ can be taken as the optimal stationary state distribution $(y^*(s, 1) + y^*(s, 0))_{s \in \mathbb{S}}$, given the optimal solution of LP, $y^*$.

The definitions of `FTVA` and `FTVA-CT` do not require any change, considering that they are already using the stationary distribution of $\bar{\pi}$ to initialize the virtual states. The conversion losses, now denoted as $V_1^{\bar{\pi},\mu} - V_N^{\text{FTVA}(\bar{\pi},\mu)}$ and $V_1^{\bar{\pi},\mu} - V_N^{\text{FTVA-CT}(\bar{\pi},\mu)}$, have the same bound as what we currently have in equation (11) in Theorem 1 for discrete-time RBs and equation (20) in Theorem 2 for continuous-time RBs. These bounds hold as long as the given input policy $\bar{\pi}$ has *finite expected synchronization times*, which can be seen by inspecting the proofs of the theorems given in Appendix E and Appendix F.

The optimality gap bounds in (12) in Theorem 1 and (21) in Theorem 2 still follow from the conversion loss bounds when the unichain assumption is dropped. To see this, for each of the discrete-time and continuous-time settings, consider an optimal single-armed policy $\bar{\pi}^*$ and optimal initial state distribution $\mu^*$. Note that $V_1^{\bar{\pi}^*,\mu^*} = V_1^{\text{rel}} \geq V_N^*$. If $V_1^{\bar{\pi}^*,\mu^*} - V_N^{\text{FTVA}(\bar{\pi}^*,\mu^*)} = O(1/\sqrt{N})$, then $V_N^* - V_N^{\text{FTVA}(\bar{\pi}^*,\mu^*)} = O(1/\sqrt{N})$; if $V_1^{\bar{\pi}^*,\mu^*} - V_N^{\text{FTVA-CT}(\bar{\pi}^*,\mu^*)} = O(1/\sqrt{N})$, then $V_N^* - V_N^{\text{FTVA-CT}(\bar{\pi}^*,\mu^*)} = O(1/\sqrt{N})$.

In the discrete-time setting, our sufficient conditions for finite expected synchronization times, presented in Propositions 4, 5, 6 and 7, are valid as long as each of the two policies $\bar{\pi}$ and $\bar{\pi}_1$ has a single recurrent class and the two classes intersect. Similarly, Lemma 5 in Appendix F, which establishes finite expected synchronization times for the continuous-time setting, holds under the same condition on $\bar{\pi}$ and $\bar{\pi}_1$.

## E   Proofs for discrete-time RBs

### E.1   Overview

In this section, we focus on proving the bound (11) on the conversion loss $V_1^{\bar{\pi}} - V_N^{\text{FTVA}(\bar{\pi})}$ for any single-armed policy $\bar{\pi}$ . The asymptotic optimality result (12) in Theorem 1 will be a direct consequence of (11) if we take $\bar{\pi}$ to be any optimal single-armed policy $\bar{\pi}^*$.

We first show that under `FTVA`$(\bar{\pi})$ defined in Algorithm 1, each virtual process is an independent Markov chain induced by applying $\bar{\pi}$ to the single-armed system $(\mathbb{S}, \mathbb{A}, P, r)$, as stated in Lemma 1 below. We will rigorously prove it in the next subsection.

**Lemma 1.** *Under* FTVA($\bar{\pi}$) *given in Algorithm 1, for each* $t = 0, 1, 2 \ldots$, $i \in [N]$, $\widehat{s} \in \mathbb{S}^N$, *and* $\widehat{a} \in \mathbb{A}^N$,

$$\mathbb{P}\left(\widehat{\boldsymbol{A}}(t) = \widehat{\boldsymbol{a}} \Big| \widehat{\boldsymbol{S}}(t), \ldots, \widehat{\boldsymbol{S}}(0)\right) = \prod_{i \in [N]} \bar{\pi}\left(\widehat{a}_i | \widehat{S}_i(t)\right) \tag{23}$$

$$\mathbb{P}\left(\widehat{\boldsymbol{S}}(t+1) = \widehat{\boldsymbol{s}} \Big| \widehat{\boldsymbol{S}}(t), \widehat{\boldsymbol{A}}(t), \ldots, \widehat{\boldsymbol{S}}(0), \widehat{\boldsymbol{A}}(0)\right) = \prod_{i \in [N]} P\left(\widehat{S}_i(t), \widehat{A}_i(t), \widehat{s}_i\right). \tag{24}$$

*Let* $(y(s,a))_{s \in \mathbb{S}, a \in \mathbb{A}}$ *be the steady-state state-action distribution of the single-armed system under* $\bar{\pi}$. *By (23)(24), for each* $t \geq 0$, $(\widehat{S}_i(t), \widehat{A}_i(t))$ *for* $i \in [N]$ *are i.i.d. with the distribution* $(y(s,a))_{s \in \mathbb{S}, a \in \mathbb{A}}$.

Next, as sketched in Section 4.2, we bound $V_1^{\bar{\pi}} - V_N^{\text{FTVA}(\bar{\pi})}$ as

$$V_1^{\bar{\pi}} - V_N^{\text{FTVA}(\bar{\pi})} = \frac{1}{N} \mathbb{E}\left[\sum_{i=1}^{N} r(\widehat{S}_i(\infty), \widehat{A}_i(\infty)) - \sum_{i=1}^{N} r(S_i(\infty), A_i(\infty))\right]$$

$$\leq \frac{2r_{\max}}{N} \mathbb{E}\left[\sum_{i=1}^{N} \mathbb{1}\left\{(\widehat{S}_i(\infty), \widehat{A}_i(\infty)) \neq (S_i(\infty), A_i(\infty))\right\}\right]. \tag{13}$$

We say an arm $i$ is a *bad arm* at time $t$ if $(\widehat{S}_i(t), \widehat{A}_i(t)) \neq (S_i(t), A_i(t))$, and a *good arm* otherwise. Then $\mathbb{E}\left[\sum_{i=1}^{N} \mathbb{1}\left\{(\widehat{S}_i(\infty), \widehat{A}_i(\infty)) \neq (S_i(\infty), A_i(\infty))\right\}\right] = \mathbb{E}[\# \text{ bad arms}]$ in steady state.

By Little's Law, we have the following relationship:

$$\mathbb{E}[\# \text{ bad arms}] = (\text{rate of generating bad arms}) \times \mathbb{E}[\text{time duration of a bad arm}].$$

To make the quantities in the above expression precise, we make the following definitions.

We say there is a *disagreement event of the arm* $i$ happening at time $t$ if $\widehat{A}_i(t) \neq A_i(t)$. The disagreement events are the only cause that turns good arms into bad arms because otherwise by the construction of our algorithm, $\widehat{S}_i(t)$ will remain the same as $S_i(t)$. The number of disagreement events in each time slot is determined by how much the budget required by virtual actions violates the constraint. We call the expected number of disagreement events when the virtual states are $\widehat{s}$ the *instantaneous disagreement rate*, denoted as $d(\widehat{s})$.

We also define *disagreement period of the arm* $i$ as a continuous period of time when arm $i$ is a bad arm, separated by disagreement events. Formally,

**Definition 2** (Disagreement period). Given a sample path of the arm $i$'s real states and virtual states $(S_i(t), \widehat{S}_i(t))_{t \geq 0}$, we define the disagreement period of the arm $i$ as a time interval $[t_{\text{begin}}, t_{\text{end}} - 1]$ such that

$$A_i(t_{\text{begin}}) \neq \widehat{A}_i(t_{\text{begin}});$$
$$A_i(t) = \widehat{A}_i(t) \text{ and } S_i(t) \neq \widehat{S}_i(t) \quad \forall t \in [t_{\text{begin}} + 1, t_{\text{end}} - 1];$$
$$A_i(t_{\text{end}}) \neq \widehat{A}_i(t_{\text{end}}) \text{ or } S_i(t_{\text{end}}) = \widehat{S}_i(t_{\text{end}}).$$

We use $D_{\text{avg}}$ to denote the long-run average length of the disagreement periods, i.e., the average of $t_{\text{end}} - t_{\text{begin}}$ of all disagreement periods.

With the definitions, we can state Little's Law in the context of disagreement periods. We omit its proof since it is a direct consequence of Little's Law.

**Lemma 2** (Little's Law for disagreement periods).

$$\mathbb{E}\left[\sum_{i=1}^{N} \mathbb{1}\left\{(\widehat{S}_i(\infty), \widehat{A}_i(\infty)) \neq (S_i(\infty), A_i(\infty))\right\}\right] = \mathbb{E}\left[d(\widehat{\boldsymbol{S}}(\infty))\right] \cdot D_{avg}, \tag{25}$$

*where* $d(\widehat{s})$ *denotes the instantaneous disagreement rate when the virtual states are* $\widehat{\boldsymbol{S}}(t) = \widehat{s}$; $D_{avg}$ *denotes the long-run average length of the disagreement periods.*

## E.2 Lemmas and the proof of Theorem 1

In this section, we will prove Theorem 1. We first rigorously prove Lemma 1.

*Proof of Lemma 1.* The first equation (23) is obvious because each virtual action $\widehat{A}_i(t)$ is sampled from $\bar{\pi}(\cdot|\widehat{S}_i(t))$ independent of anything else. We only need to show (24). Recall that by the definition of $\mathtt{FTVA}(\bar{\pi})$, $\widehat{S}_i(t+1)$ could either be equal to $S_i(t+1)$ if $(\widehat{S}_i(t), \widehat{A}_i(t)) = (S_i(t), A_i(t))$, or generated afresh independent of anything else if $(\widehat{S}_i(t), \widehat{A}_i(t)) \neq (S_i(t), A_i(t))$. Let $\mathcal{I}_{\mathrm{good}}(t) = \{i : (\widehat{S}_i(t), \widehat{A}_i(t)) = (S_i(t), A_i(t))\}$. Then

$$
\begin{aligned}
&\mathbb{P}\left(\widehat{\boldsymbol{S}}(t+1) = \widehat{\boldsymbol{s}} \,\Big|\, \widehat{\boldsymbol{S}}(t), \widehat{\boldsymbol{A}}(t), \ldots, \widehat{\boldsymbol{S}}(0), \widehat{\boldsymbol{A}}(0)\right) \\
&= \prod_{i \in \mathcal{I}_{\mathrm{good}}(t)} \mathbb{P}\left(S_i(t+1) = \widehat{s}_i \,\Big|\, S_i(t), A_i(t)\right) \prod_{i \notin \mathcal{I}_{\mathrm{good}}(t)} \mathbb{P}\left(\widehat{S}_i(t+1) = \widehat{s}_i \,\Big|\, \widehat{S}_i(t), \widehat{A}_i(t)\right) \\
&= \prod_{i \in \mathcal{I}_{\mathrm{good}}(t)} P\left(S_i(t), A_i(t), \widehat{s}_i\right) \prod_{i \notin \mathcal{I}_{\mathrm{good}}(t)} P\left(\widehat{S}_i(t), \widehat{A}_i(t), \widehat{s}_i\right) \\
&= \prod_{i \in [N]} P\left(\widehat{S}_i(t), \widehat{A}_i(t), \widehat{s}_i\right),
\end{aligned}
$$

where the last equality is by the definition of $\mathcal{I}_{\mathrm{good}}(t)$.

Combining (23) and (24), we have confirmed that each virtual process is an independent single-armed process with transition kernel $P$ and policy $\bar{\pi}$. Because the initial virtual states $\{\widehat{S}_i(0)\}_{i \in [N]}$ are sampled i.i.d. from the stationary distribution of $\bar{\pi}$, $\{\widehat{S}_i(t)\}_{i \in [N]}$ are also i.i.d. with the same distribution. Therefore, $\{(\widehat{S}_i(t), \widehat{A}_i(t))\}_{i \in [N]}$ are i.i.d. with distribution $(y(s,a))_{s \in \mathbb{S}, a \in \mathbb{A}}$. □

The next two lemmas bound the two quantities in Lemma 2, the long-run average disagreement rate and the long-run average length of the disagreement periods.

**Lemma 3** (Average disagreement rates)**.** *The instantaneous disagreement rate is equal to*

$$
d(\widehat{\boldsymbol{s}}) = \mathbb{E}\left[\left|\sum_{i=1}^{N} \widehat{A}_i(t) - \alpha N\right| \;\Big|\; \widehat{\boldsymbol{S}}(t) = \widehat{\boldsymbol{s}}\right]. \tag{26}
$$

*The long-run average disagreement rate is bounded as*

$$
\mathbb{E}\left[d(\widehat{\boldsymbol{S}}(\infty))\right] \leq \frac{1}{2}\sqrt{N}. \tag{27}
$$

*Proof.* We first prove the expression of instantaneous disagreement rate in (26). By definition, the instantaneous disagreement rate is equal to the expected number of arms such that $A_i(t) \neq \widehat{A}_i(t)$ conditioning on the virtual states being $\widehat{\boldsymbol{s}}$. Because $\mathtt{FTVA}$ tries to match as many $A_i(t)$'s to $\widehat{A}_i(t)'s$ as possible, there are exactly $|\sum_{i=1}^{N} \widehat{A}_i(t) - \alpha N|$ arms such that $A_i(t) \neq \widehat{A}_i(t)$. This proves (26).

To bound the average disagreement rate, letting $\widehat{\boldsymbol{s}} = \widehat{\boldsymbol{S}}(\infty)$ in (26) and taking expectation, we have

$$
\begin{aligned}
\mathbb{E}\left[d(\widehat{\boldsymbol{S}}(\infty))\right] &= \mathbb{E}\left[\left|\sum_{i=1}^{N} \widehat{A}_i(\infty) - \alpha N\right|\right] \\
&\leq \left(\mathbb{E}\left[\left(\sum_{i=1}^{N} \widehat{A}_i(\infty) - \alpha N\right)^2\right]\right)^{1/2}. 
\end{aligned} \tag{28}
$$

By Lemma 1, the virtual actions $\widehat{A}_i(\infty)$'s are i.i.d. binomial random variables such that $\mathbb{P}(\widehat{A}_i(\infty) = 1) = \sum_{s \in \mathbb{S}} y(s,1) = \alpha$, $\sum_{i=1}^{N} \widehat{A}_i(\infty)$ has distribution Binomial$(N, \alpha)$, whose mean and variance are $\alpha N$ and $N\alpha(1-\alpha) \leq N/4$. Therefore the expression in (28) is equal to $\mathrm{Var}\left[\sum_{i=1}^{N} \widehat{A}_i(\infty)\right]^{1/2} = \sqrt{N}/2$. □

**Lemma 4** (Average length of disagreement periods)**.**

$$D_{avg} \leq \overline{\tau}_{\max}^{\text{sync}}. \tag{29}$$

*Proof.* To bound $D_{\text{avg}}$, it suffices to bound the expected length of a disagreement period with arbitrary initial states. Without loss of generality, consider a disagreement period on arm $i$ that starts at time $t_{\text{begin}} = 0$, with initial states $(S_i(0), A_i(0), \widehat{S}_i(0), \widehat{A}_i(0)) = (s, a, \widehat{s}, \widehat{a})$. During the disagreement period, the $i$-th arm can be seen as a leader-and-follower system, where the real state-action pair $(S_i(t), A_i(t))$ corresponds to the follower arm and the virtual state-action pair $(\widehat{S}_i(t), \widehat{A}_i(t))$ corresponds to the leader arm. By Assumption 1, the leader arm and the follower arm will synchronize in a finite expected time. When they synchronize, the disagreement period stops, which means $t_{\text{end}} \leq \tau^{\text{sync}}(s, a, \widehat{s}, \widehat{a})$. Therefore, the expected length of the disagreement period satisfies

$$\mathbb{E}\left[t_{\text{end}} - t_{\text{begin}}\right] = \mathbb{E}\left[t_{\text{end}}\right] \leq \mathbb{E}\left[\tau^{\text{sync}}(s, a, \widehat{s}, \widehat{a})\right] \leq \overline{\tau}_{\max}^{\text{sync}}.$$

This holds for arbitrary initial states $(s, a, \widehat{s}, \widehat{a})$, so $D_{\text{avg}} \leq \overline{\tau}_{\max}^{\text{sync}}$. $\qquad\square$

Given the three lemmas above, we can prove Theorem 1.

*Proof of Theorem 1.* Combining Lemma 2, 4 and 3, we have

$$\mathbb{E}\left[\sum_{i=1}^{N} \mathbb{1}\left\{(\widehat{S}_i(\infty), \widehat{A}_i(\infty)) \neq (S_i(\infty), A_i(\infty))\right\}\right] \leq \frac{1}{2}\overline{\tau}_{\max}^{\text{sync}}\sqrt{N}.$$

Plugging the above inequality into the bound on conversion loss in (13), we get

$$
\begin{aligned}
V_1^{\bar{\pi}} - V_N^{\texttt{FTVA}(\bar{\pi})} &= \frac{1}{N}\mathbb{E}\left[\sum_{i=1}^{N} r(\widehat{S}_i(\infty), \widehat{A}_i(\infty)) - \sum_{i=1}^{N} r(S_i(\infty), A_i(\infty))\right] \\
&\leq \frac{2r_{\max}}{N}\mathbb{E}\left[\sum_{i=1}^{N} \mathbb{1}\left\{(\widehat{S}_i(\infty), \widehat{A}_i(\infty)) \neq (S_i(\infty), A_i(\infty))\right\}\right] \\
&\leq \frac{r_{\max}\overline{\tau}_{\max}^{\text{sync}}}{\sqrt{N}}.
\end{aligned}
$$

This finishes the proof. $\qquad\square$

## F Proofs for continuous-time RBs

### F.1 Preliminary: finite synchronization time

In this section, we prove that the synchronization time has a finite expectation in the continuous-time setting. Unlike the discrete-time setting, we do not need to make any additional assumptions other than the standard unichain condition. Our proof is based on the observation that the holding time distribution of a continuous-time Markov chain has support on the whole positive real line, so an arbitrary number of transitions can happen in any time interval, which, from the uniformization perspective, implies self-loops in all states. We can thus prove that synchronization time has a finite expectation using similar logic as in the proof of Proposition 3 and 4.

**Lemma 5** (Synchronization in continuous time)**.** *Consider the single-armed policy $\bar{\pi}$ and the corresponding leader-and-follower system. Given the initial states $(S(0), \widehat{S}(0)) = (s, \widehat{s})$, the synchronization time is bounded as*

$$\mathbb{E}\left[\tau^{\text{sync}}(s, \widehat{s})\right] < \infty.$$

*Proof of Lemma 5.* Consider the always-1 policy $\bar{\pi}_1$ given by

$$\bar{\pi}_1(a|s) = \begin{cases} 1, & \text{if } a = 1, \\ 0, & \text{if } a = 0. \end{cases} \quad \text{for } s \in \mathbb{S}, a \in \mathbb{A}.$$

By the unichain property, $\bar{\pi}_1$ only has a single recurrent class, which we denote as $\mathcal{S}_1$. We denote the recurrent class under $\bar{\pi}$ as $\mathcal{S}$.

We first prove by contradiction that $\mathcal{S} \cap \mathcal{S}_1 \neq \emptyset$. Suppose $\mathcal{S} \cap \mathcal{S}_1 = \emptyset$. Then we define a new policy $\bar{\pi}'$ as

$$\bar{\pi}'(a|s) = \begin{cases} \bar{\pi}(a|s), & \text{if } s \in \mathcal{S}, \\ \bar{\pi}_1(a|s), & \text{otherwise.} \end{cases} \quad \text{for } s \in \mathbb{S}, a \in \mathbb{A}.$$

Then $\mathcal{S}$ and $\mathcal{S}_1$ are two distinct recurrent classes under $\bar{\pi}'$. This is because by definition of $\bar{\pi}'$, an arm with the initial state in $\mathcal{S}$ remains in $\mathcal{S}$, so it cannot reach $\mathcal{S}_1$; similarly an arm cannot reach $\mathcal{S}$ from $\mathcal{S}_1$. The existence of two recurrent classes contradicts the unichain condition. Therefore, we must have $\mathcal{S} \cap \mathcal{S}_1 \neq \emptyset$.

We show that given any pair of initial states $s, \widehat{s} \in \mathbb{S}$, the probability that $\tau^{\text{sync}}(s, \widehat{s}) < 3$ is positive. We construct a sequence of positive probability events that leads to $S(t) = \widehat{S}(t)$ before time 3:

1. The leader arm reaches a state $s^a$ such that $\bar{\pi}(1|s^a) > 0$ by time 1 and chooses action 1;

2. The leader arm has no transition during $[1, 2]$, so $\widehat{S}(t) = s^a$ and $\widehat{A}(t) = 1$ for all $t \in [1, 2]$;

3. The follower arm applies the same action as the leader arm during $[1, 2]$, so $A(t) = 1$ for all $t \in [1, 2]$; and reaches the state $s^b$ by time 2 for some $s^b \in \mathcal{S} \cap \mathcal{S}_1$;

4. The follower arm stays at $s^b$ during $[2, 3]$, and the leader arm reaches $s^b$ by time 3, so the two arms synchronize.

The first two events have positive probabilities because the policy $\bar{\pi}$ applies action 1 with a positive fraction of time. To see why the third event has a positive probability, observe that the follower arm applies action 1 for all $t \in [1, 2]$ regardless of its state, so it is effectively under policy $\bar{\pi}_1$ and could traverse all states in $\mathcal{S}_1$. The fourth event has a positive probability because the leader arm under $\bar{\pi}$ can traverse all states in $\mathcal{S}$. Note that in the above arguments, we are implicitly assuming that the two arms do not synchronize in the middle of the four events, because otherwise we are done. Also, we use the fact that in a continuous-time Markov chain, there can be an arbitrary number of transitions during any time interval. Therefore, we have proved that for all $s, \widehat{s} \in \mathbb{S}$,

$$\mathbb{P}\left(\tau^{\text{sync}}(s, \widehat{s}) < 3\right) > 0.$$

Now we prove by induction that for all $s, \widehat{s} \in \mathbb{S}$ and $k = 0, 1, 2, \ldots$,

$$\mathbb{P}\left(\tau^{\text{sync}}(s, \widehat{s}) \geq 3k\right) \leq \left(\max_{s', \widehat{s}' \in \mathbb{S}} \mathbb{P}\left(\tau^{\text{sync}}(s', \widehat{s}') \geq 3\right)\right)^k. \tag{30}$$

The base case of $k = 1$ is already known. Suppose we have proved (30) for a certain $k$, then

$$\mathbb{P}\left(\tau^{\text{sync}}(s, \widehat{s}) \geq 3(k+1)\right) = \mathbb{P}\left(\tau^{\text{sync}}(s, \widehat{s}) \geq 3k\right) \cdot \mathbb{P}\left(\tau^{\text{sync}}(s, \widehat{s}) \geq 3(k+1) \mid \tau^{\text{sync}}(s, \widehat{s}) \geq 3k\right)$$

$$= \mathbb{P}\left(\tau^{\text{sync}}(s, \widehat{s}) \geq 3k\right) \cdot \mathbb{P}\left(\tau^{\text{sync}}(S(3k), \widehat{S}(3k)) \geq 3 \mid S(3k) \neq \widehat{S}(3k)\right)$$

$$\leq \mathbb{P}\left(\tau^{\text{sync}}(s, \widehat{s}) \geq 3k\right) \cdot \max_{s', \widehat{s}' \in \mathbb{S}} \mathbb{P}\left(\tau^{\text{sync}}(s', \widehat{s}') \geq 3\right)$$

$$\leq \left(\max_{s', \widehat{s}' \in \mathbb{S}} \mathbb{P}\left(\tau^{\text{sync}}(s', \widehat{s}') \geq 3\right)\right)^{k+1}.$$

where in the second equality we have used the Markov property of the system, and in the last inequality we apply the induction hypothesis. This proves (30).

The bound on the expectation $\mathbb{E}\left[\tau^{\text{sync}}(s, \widehat{s})\right]$ follows summing the tail bound (30) over $k = 0, 1, 2, \ldots$:

$$\mathbb{E}\left[\tau^{\text{sync}}(s, \widehat{s})\right] \leq \sum_{k=0}^{\infty} 3\mathbb{P}\left(\tau^{\text{sync}}(s, \widehat{s}) \geq 3k\right)$$

$$\leq \frac{3}{1 - \max_{s', \widehat{s}' \in \mathbb{S}} \mathbb{P}\left(\tau^{\text{sync}}(s', \widehat{s}') \geq 3\right)}$$

$$< \infty,$$

where the second inequality is due to (30) and the fact that $\max_{s', \widehat{s}' \in \mathbb{S}} \mathbb{P}\left(\tau^{\text{sync}}(s', \widehat{s}') \geq 3\right) \leq 1 - \min_{s', \widehat{s}' \in \mathbb{S}} \mathbb{P}\left(\tau^{\text{sync}}(s', \widehat{s}') < 3\right) < 1$. This finishes the proof. $\qquad \square$

## F.2 Overview of the proof of Theorem 2

In this section, we focus on proving the bound (20) on the conversion loss $V_1^{\bar{\pi}} - V_N^{\texttt{FTVA-CT}(\bar{\pi})}$ for any single-armed policy $\bar{\pi}$. The asymptotic optimality result (21) in Theorem 2 will be a direct consequence of (20) if we take $\bar{\pi}$ to be any optimal single-armed policy $\bar{\pi}^*$.

We first show that under $\texttt{FTVA-CT}(\bar{\pi})$ defined in Algorithm 2, $\{(\boldsymbol{S}(t), \widehat{\boldsymbol{S}}(t))\}_{t \geq 0}$ is a continuous-time Markov chain and each virtual process is an independent Markov chain induced by applying $\bar{\pi}$ to the single-armed system, as stated in Lemma 6 below. We will prove it in the next subsection.

**Lemma 6.** *Under the $\texttt{FTVA-CT}(\bar{\pi})$ defined in Algorithm 2, we have the following:*

*(1)* $\{(\boldsymbol{S}(t), \widehat{\boldsymbol{S}}(t))\}_{t \geq 0}$ *is a continuous-time Markov chain.*

*(2)* $\{\widehat{S}_i(t)\}_{t \geq 0}$ *for $i \in [N]$ are $N$ independent Markov chains, whose transition rate from state $s$ to $s'$ is $\sum_{a \in \mathbb{A}} G(s, a, s') \bar{\pi}(a|s)$, for $s, s' \in \mathbb{S}$ s.t. $s \neq s'$.*

*Let $(y(s, a))_{s \in \mathbb{S}, a \in \mathbb{A}}$ be the steady-state state-action distribution of the single-armed system under $\bar{\pi}$. Then the second result above implies that for each $t \geq 0$, $(\widehat{S}_i(t), \widehat{A}_i(t))$ for $i \in [N]$ are i.i.d. with the distribution $(y(s, a))_{s \in \mathbb{S}, a \in \mathbb{A}}$.*

Next, we can upper bound the conversion loss $V_1^{\bar{\pi}} - V_N^{\texttt{FTVA-CT}(\bar{\pi})}$ as

$$
\begin{aligned}
V_1^{\bar{\pi}} - V_N^{\texttt{FTVA-CT}(\bar{\pi})} &= \frac{1}{N} \mathbb{E} \left[ \sum_{i=1}^{N} r(\widehat{S}_i(\infty), \widehat{A}_i(\infty)) - \sum_{i=1}^{N} r(S_i(\infty), A_i(\infty)) \right] \\
&\leq \frac{2 r_{\max}}{N} \mathbb{E} \left[ \sum_{i=1}^{N} \mathbb{1} \left\{ (\widehat{S}_i(\infty), \widehat{A}_i(\infty)) \neq (S_i(\infty), A_i(\infty)) \right\} \right] \\
&\leq \frac{2 r_{\max}}{N} \mathbb{E} \left[ \sum_{i=1}^{N} \mathbb{1} \left\{ \widehat{S}_i(\infty) \neq S_i(\infty) \right\} + \sum_{i=1}^{N} \mathbb{1} \left\{ \widehat{A}_i(\infty) \neq A_i(\infty) \right\} \right]. \quad (31)
\end{aligned}
$$

The bound is slightly different from the (13) since we want to separately deal with the number of arms whose real and virtual states do not agree and those whose real and virtual actions do not agree, as shown in the lemma below.

**Lemma 7.** *It holds that*

$$
\mathbb{E} \left[ \sum_{i=1}^{N} \mathbb{1} \left\{ \widehat{S}_i(\infty) \neq S_i(\infty) \right\} \right] \leq 2 g_{\max} \bar{\tau}_{\max}^{\texttt{sync}} \mathbb{E} \left[ \left| \sum_{i=1}^{N} \widehat{A}_i(\infty) - \alpha N \right| \right] \quad (32)
$$

$$
\mathbb{E} \left[ \sum_{i=1}^{N} \mathbb{1} \left\{ \widehat{A}_i(\infty) \neq A_i(\infty) \right\} \right] = \mathbb{E} \left[ \left| \sum_{i=1}^{N} \widehat{A}_i(\infty) - \alpha N \right| \right] \quad (33)
$$

Before showing the proof of Lemma 7, we will first use Lemma 7 to prove Theorem 2.

*Proof of Theorem 2.* Combining (31) and Lemma 7, we have

$$
V_1^{\bar{\pi}} - V_N^{\texttt{FTVA-CT}(\bar{\pi})} \leq \frac{2 r_{\max}}{N} (1 + 2 g_{\max} \bar{\tau}_{\max}^{\texttt{sync}}) \mathbb{E} \left[ \left| \sum_{i=1}^{N} \widehat{A}_i(\infty) - \alpha N \right| \right].
$$

By Lemma 6, the virtual actions $\widehat{A}_i(\infty)$'s are i.i.d. binomial random variables such that $\mathbb{P}(\widehat{A}_i(\infty) = 1) = \sum_{s \in \mathbb{S}} y(s, a) = \alpha$, so the distribution of $\sum_{i=1}^{N} \widehat{A}_i(\infty)$ is Binomial$(N, \alpha)$, whose mean and variance are $\alpha N$ and $N\alpha(1 - \alpha) \leq N/4$. Therefore,

$$
\mathbb{E} \left[ \left| \sum_{i=1}^{N} \widehat{A}_i(\infty) - \alpha N \right| \right] \leq \left( \mathbb{E} \left[ \left( \sum_{i=1}^{N} \widehat{A}_i(\infty) - \alpha N \right)^2 \right] \right)^{1/2}
$$

$$= \text{Var}\left[\sum_{i=1}^{N} \widehat{A}_i(\infty)\right]^{1/2}$$

$$\leq \frac{1}{2}\sqrt{N}.$$

Combining the above calculations, we get

$$V_1^{\bar{\pi}} - V_N^{\text{FTVA-CT}(\bar{\pi})} \leq \frac{r_{\max}(1 + 2g_{\max}\bar{\tau}_{\max}^{\text{sync}})}{\sqrt{N}}.$$

$\square$

### F.3 Proof of Lemma 6

*Proof.* We first show that $\{(\boldsymbol{S}(t), \widehat{\boldsymbol{S}}(t))\}_{t \geq 0}$ is a continuous-time Markov chain. Observe that $\{(\boldsymbol{S}(t), \widehat{\boldsymbol{S}}(t))\}_{t \geq 0}$ is piecewise constant between decision epochs $\{t_k\}_{k=0,1,\dots}$, so it suffices to consider the time between the decision epochs $t_{k+1} - t_k$ and the states at the decision epochs $(\boldsymbol{S}(t_k), \widehat{\boldsymbol{S}}(t_k))$ for $k = 0, 1, 2, \dots$

We claim that at each decision epoch $t_k$, the time until the next decision epoch $t_{k+1} - t_k$ is exponentially distributed conditioned on $(\boldsymbol{S}(t_k), \widehat{\boldsymbol{S}}(t_k))$. Observe from the pseudo-code in Algorithm 2 that at the decision epoch $t_k$, conditioned on $(\boldsymbol{S}(t_k), \boldsymbol{A}(t_k), \widehat{\boldsymbol{S}}(t_k))$, the time until the next decision epoch $t_{k+1}$ is the minimum of two independent exponential random variables, corresponding to the exponential timer and the transition of real states. The rates of the two exponential random variables are $2Ng_{\max} - g_k^{\text{real}}$ and $g_k^{\text{real}}$, where $g_k^{\text{real}} = \sum_{i=1}^{N} G(S_i(t_k), A_i(t_k))$. Therefore, conditioned on $(\boldsymbol{S}(t_k), \boldsymbol{A}(t_k), \widehat{\boldsymbol{S}}(t_k))$, $t_{k+1} - t_k$ has an exponential distribution with rate $2Ng_{\max} - g_k^{\text{real}} + g_k^{\text{real}} = 2Ng_{\max}$. Because the rate $2Ng_{\max}$ is a constant, if we take expectation over $\boldsymbol{A}(t_k)$ and only conditioned on $(\boldsymbol{S}(t_k), \widehat{\boldsymbol{S}}(t_k))$, the time until the next decision epoch $t_{k+1} - t_k$ is still exponentially distributed with rate $2Ng_{\max}$.

Also, it is obvious from the pseudo-code in Algorithm 2 that conditioned on $(\boldsymbol{S}(t_k), \widehat{\boldsymbol{S}}(t_k))$, the distribution of $(\boldsymbol{S}(t_{k+1}), \widehat{\boldsymbol{S}}(t_{k+1}))$ is independent of $t_{k+1} - t_k$ and $(\boldsymbol{S}(t), \widehat{\boldsymbol{S}}(t))$ for $t < t_k$. Therefore, we conclude that $\{(\boldsymbol{S}(t), \widehat{\boldsymbol{S}}(t))\}_{t \geq 0}$ is a continuous-time Markov chain.

Next, we show that the $\{\widehat{S}_i(t)\}_{t \geq 0}$ for $i \in [N]$ are $N$ independent Markov chains, whose transition rate from state $s$ to $s'$ is $\sum_{a \in \mathbb{A}} G(s, a, s')\bar{\pi}(a|s)$ for $s' \neq s$. Because we have shown that $\{(\boldsymbol{S}(t), \widehat{\boldsymbol{S}}(t))\}_{t \geq 0}$ is a continuous-time Markov chain with a constant transition rate, it suffices to focus on the embedded chain $\{(\boldsymbol{S}(t_k), \widehat{\boldsymbol{S}}(t_k))\}_{k=0,1,\dots}$ and examine the probability of the transitions that change the virtual states $\widehat{\boldsymbol{S}}(t_k)$. Note that between two decision epochs $t_k$ and $t_{k+1}$, there is at most one $i$ such that $\widehat{S}_i(t)$ changes. Let $\mathcal{I}_{\text{good}}(t_k) = \{i \in [N] \colon \widehat{S}_i(t_k) = S_i(t_k), \widehat{A}_i(t_k) = A_i(t_k)\}$. For any arm $i \in \mathcal{I}_{\text{good}}(t_k)$, the transitions of its virtual and real states are coupled, which implies that the probability that $\widehat{S}_i(t_{k+1}) = s'$ is the same as the probability that $S_i(t_{k+1}) = s'$, for any $s' \neq \widehat{S}_i(t_k)$, conditioned on the states and actions at $t_k$. Formally,

$$\mathbb{P}\Big(\widehat{S}_i(t_{k+1}) = s' \Big| \boldsymbol{S}(t_k), \boldsymbol{A}(t_k), \widehat{\boldsymbol{S}}(t_k), \widehat{\boldsymbol{A}}(t_k)\Big) \mathbb{1}\{i \in \mathcal{I}_{\text{good}}(t_k)\}$$

$$= \mathbb{P}\Big(S_i(t_{k+1}) = s' \Big| \boldsymbol{S}(t_k), \boldsymbol{A}(t_k), \widehat{\boldsymbol{S}}(t_k), \widehat{\boldsymbol{A}}(t_k)\Big) \mathbb{1}\{i \in \mathcal{I}_{\text{good}}(t_k)\}$$

$$= \frac{G(S_i(t_k), A_i(t_k), s')}{2Ng_{\max}} \mathbb{1}\{i \in \mathcal{I}_{\text{good}}(t_k)\}$$

$$= \frac{G(\widehat{S}_i(t_k), \widehat{A}_i(t_k), s')}{2Ng_{\max}} \mathbb{1}\{i \in \mathcal{I}_{\text{good}}(t_k)\}, \tag{34}$$

where the second equality uses the fact that the transition probability after uniformization is equal to the ratio between the transition rate and the uniformization rate; the last equality is by the definition of $\mathcal{I}_{\text{good}}(t_k)$. For $i \notin \mathcal{I}_{\text{good}}(t_k)$ and $s' \neq \widehat{S}_i(t_k)$, we have $\widehat{S}_i(t_{k+1}) = s'$ only when the exponential

clock ticks and the pair $(i, s')$ is sampled, so

$$
\mathbb{P}\Big(\widehat{S}_i(t_{k+1}) = s' \Big| \boldsymbol{S}(t_k), \boldsymbol{A}(t_k), \widehat{\boldsymbol{S}}(t_k), \widehat{\boldsymbol{A}}(t_k)\Big) \mathbb{1}\{i \notin \mathcal{I}_{\text{good}}(t_k)\}
$$

$$
= \mathbb{E}\left[\frac{2N g_{\max} - g_k^{\text{real}}}{2N g_{\max}} \frac{G(\widehat{S}_i(t_k), \widehat{A}_i(t_k), s')}{2N g_{\max} - g_k^{\text{real}}} \Bigg| \boldsymbol{S}(t_k), \boldsymbol{A}(t_k), \widehat{\boldsymbol{S}}(t_k), \widehat{\boldsymbol{A}}(t_k)\right] \mathbb{1}\{i \notin \mathcal{I}_{\text{good}}(t_k)\}
$$

$$
= \frac{G(\widehat{S}_i(t_k), \widehat{A}_i(t_k), s')}{2N g_{\max}} \mathbb{1}\{i \notin \mathcal{I}_{\text{good}}(t_k)\}. \tag{35}
$$

Summing up (34)(35), we have that for any $i \in [N]$ and $s' \neq \widehat{S}_i(t_k)$,

$$
\mathbb{P}\Big(\widehat{S}_i(t_{k+1}) = s' \Big| \boldsymbol{S}(t_k), \boldsymbol{A}(t_k), \widehat{\boldsymbol{S}}(t_k), \widehat{\boldsymbol{A}}(t_k)\Big) = \frac{G(\widehat{S}_i(t_k), \widehat{A}_i(t_k), s')}{2N g_{\max}}. \tag{36}
$$

Because $\widehat{A}_i(t_k)$ is independently sampled from the distribution $\bar{\pi}(\cdot|\widehat{S}_i(t_k))$, taking expectation over $(\boldsymbol{A}(t_k), \widehat{\boldsymbol{A}}(t_k))$ in the above equation, we get

$$
\mathbb{P}\Big(\widehat{S}_i(t_{k+1}) = s' \Big| \boldsymbol{S}(t_k), \widehat{\boldsymbol{S}}(t_k)\Big) = \frac{\sum_{a \in \mathbb{A}} G(\widehat{S}_i(t_k), a, s')\bar{\pi}\big(a|\widehat{S}_i(t_k)\big)}{2N g_{\max}}, \tag{37}
$$

for any $i \in [N]$ and $s' \neq \widehat{S}_i(t_k)$. Because the uniformization rate is $2N g_{\max}$, the rate for $\widehat{S}_i(t)$ to transition to $s'$ is equal to $\sum_{a \in \mathbb{A}} G(\widehat{S}_i(t), a, s')\bar{\pi}\big(a|\widehat{S}_i(t)\big)$ conditioned on $(\boldsymbol{S}(t), \widehat{\boldsymbol{S}}(t))$. Observe that this transition rate only depends on $\widehat{S}_i(t)$, which implies that $\widehat{S}_i(t)$'s for $i \in [N]$ are $N$ i.i.d. Markov chains.

Finally, recall that we initialize the virtual states $\widehat{S}_i(0)$'s as $N$ i.i.d. samples from the stationary distribution of the single-armed system under the policy $\bar{\pi}$. Since we have proved that $\{\widehat{S}_i(t)\}_{t \geq 0}$'s are $N$ i.i.d. Markov chains induced by applying $\bar{\pi}$ to the single-armed systems, for each $t \geq 0$, $\widehat{S}_i(t)$'s remain stationary and i.i.d. Therefore, for each $t \geq 0$, $(\widehat{S}_i(t), \widehat{A}_i(t))$'s are i.i.d., and for each $i$, the distribution of $(\widehat{S}_i(t), \widehat{A}_i(t))$ is equal to the steady-state state-action distribution $(y(s, a))_{s \in \mathbb{S}, a \in \mathbb{A}}$. $\qquad\square$

### F.4 Proof of Lemma 7

In this section, we prove the key intermediate result, Lemma 7. The second equation (33) in Lemma 7 is obvious from the definition of the policy. We can therefore focus on proving (32), i.e., bounding the long-run average number of arms whose real states are not equal to their virtual states. We call such arms *bad arms*, and the rest of the arms *good arms*.

We will use a similar approach as Section E: we invoke Little's Law to write the average number of bad arms as a product of the average *disagreement rate* and the average length of *disagreement periods*, based on a suitable definition of the *disagreement events*.

We define the disagreement event in a different way than in the discrete-time RB setting, because unlike in the discrete-time RB setting where $A_i(t) \neq \widehat{A}_i(t)$ can immediately cause $S_i(t) \neq \widehat{S}_i(t)$, in the continuous-time RB setting, actions that last for only a short time may not have an effect on the states. Therefore, we say a disagreement event for arm $i$ happens at time $t$ only when arm $i$ behaves differently than it would have behaved if $A_i(t) = \widehat{A}_i(t)$.

Specifically, the disagreement event of a good arm can be defined as having a state transition that turns it into a bad arm, since a good arm is supposed to remain good if we always have $A_i(t) = \widehat{A}_i(t)$; for bad arms, how the arm "would have behaved" need to be defined with the help of an extra structure, namely, exponential timers, as described below.

**Definition 3** (Exponential timers for simulating the leader-and-follower system)**.** For each $i \in [N]$, we run a timer for each arm $i$ that ticks every random amount of time. When the timer of arm $i$ starts, it decides the time of its next tick based on the $i$-th arm $(S_i(t), A_i(t), \widehat{S}_i(t), \widehat{A}_i(t))$: if $S_i(t) \neq \widehat{S}_i(t)$ *and* $A_i(t) \neq \widehat{A}_i(t)$, the timer ticks after a time that is exponentially distributed with rate $G(S_i(t), A_i(t))$; otherwise, the timer pauses. The timer restarts when it ticks or when there is any event happening in the system.

The purpose of the exponential timer is to simulate the transition time of the leader-and-follower system, which characterizes how the arm would have behaved if $A_i(t) = \widehat{A}_i(t)$. Specifically, when arm $i$ has $S_i(t) \neq \widehat{S}_i(t)$ (so it is a bad arm) and $A_i(t) \neq \widehat{A}_i(t)$, we construct an imaginary leader-and-follower system whose state-action pairs are $(S_i(t), \widehat{A}_i(t), \widehat{S}_i(t), \widehat{A}_i(t))$. We let the transition times of the virtual state $\widehat{S}_i(t)$ in the two systems be identical, and the transition times of the real state $S_i(t)$ in the two systems be independent. Then the transition time of $S_i(t)$ in the leader-and-follower system has an exponential distribution with rate $G(S_i(t), \widehat{A}_i(t))$, which is equal to the time that the exponential timer ticks.

Therefore, there are two ways that the transition of arm $i$ with $S_i(t) \neq \widehat{S}_i(t)$ and $A_i(t) \neq \widehat{A}_i(t)$ can deviate from the leader-and-follower system described above: either arm $i$ itself has a transition in the real state $S_i(t)$, or when the exponential timer ticks. This statement is actually also true if $S_i(t) = \widehat{S}_i(t)$ and $A_i(t) \neq \widehat{A}_i(t)$ because in that case, the exponential timer pauses.

We can thus formally define the *disagreement event of arm $i$* as below.

**Definition 4** (Disagreement event for continuous-time RBs). For each $i \in [N]$, a disagreement event of arm $i$ happens when its real state $S_i(t)$ transitions or its exponential timer ticks, while $A_i(t) \neq \widehat{A}_i(t)$.

After defining the disagreement events, we can define the *disagreement period of arm $i$* as the period of time when $S_i(t) \neq \widehat{S}_i(t)$, separated by disagreement events, formally stated below.

**Definition 5** (Disagreement period for continuous-time RBs). Given a sample path of the arm $i$'s real states and virtual states $(S_i(t), \widehat{S}_i(t))$, and an exponential timer, we define the disagreement period of the arm $i$ as a time interval $[t_{\text{begin}}, t_{\text{end}})$ such that

There is a disagreement event at $t_{\text{begin}}$;

There is no disagreement event during $(t_{\text{begin}}, t_{\text{end}})$, and $S_i(t) \neq S_i(t)$ for $t \in (t_{\text{begin}}, t_{\text{end}})$;

There is a disagreement event at $t_{\text{end}}$ or $S_i(t_{\text{end}}) = S_i(t_{\text{end}})$.

We let $d(\boldsymbol{s}, \widehat{\boldsymbol{s}})$ be the *instantaneous rate of disagreement events* (instantaneous disagreement rate) when the system has real and virtual states $(\boldsymbol{S}(t), \widehat{\boldsymbol{S}}(t)) = (\boldsymbol{s}, \widehat{\boldsymbol{s}})$. Let $D_{\text{avg}}$ be the long-run average length of the disagreement periods.

Observe that the number of bad arms (the arms such that $S_i(t) \neq \widehat{S}_i(t)$) is the number of arms in disagreement periods, so we can apply Little's Law to the number of bad arms.

**Lemma 8** (Little's Law for disagreement periods, continuous-time version). *It holds that*

$$\mathbb{E}\left[\sum_{i=1}^{N} \mathbb{1}\left\{\widehat{S}_i(\infty) \neq S_i(\infty)\right\}\right] = \mathbb{E}\left[d(\boldsymbol{S}(\infty), \widehat{\boldsymbol{S}}(\infty))\right] \cdot D_{avg}, \tag{38}$$

*where $d(\boldsymbol{s}, \widehat{\boldsymbol{s}})$ denotes the instantaneous disagreement rate when the virtual states are $\widehat{\boldsymbol{S}}(t) = \widehat{\boldsymbol{s}}$; $D_{avg}$ denotes the long-run average length of the disagreement periods.*

**Lemma 9** (Average disagreement rates). *The instantaneous disagreement rate is equal to*

$$d(\boldsymbol{s}, \widehat{\boldsymbol{s}}) \leq 2g_{\max}\mathbb{E}\left[\left|\sum_{i=1}^{N} \widehat{A}_i(t) - \alpha N\right| \mid \widehat{\boldsymbol{S}}(t) = \widehat{\boldsymbol{s}}\right]. \tag{39}$$

*Proof.* For each arm $i$ such that $A_i(t) \neq \widehat{A}_i(t)$, a disagreement event happens if its real state transitions, which happens at the rate $G(S_i(t), A_i(t))$, or if its exponential timer ticks, which happens at the rate $0$ (if it is a good arm) or $G(S_i(t), \widehat{A}_i(t))$ (if it is a bad arm). Therefore, the rate that disagreement events happen at arm $i$ is no more than

$$G(S_i(t), A_i(t)) + G(S_i(t), \widehat{A}_i(t)) \leq 2g_{\max}.$$

By the definition of the policy, there are in expectation $\mathbb{E}\left[\left|\sum_{i=1}^{N} \widehat{A}_i(t) - \alpha N\right| \mid \widehat{\boldsymbol{S}}(t) = \widehat{\boldsymbol{s}}\right]$ arms with $A_i(t) \neq \widehat{A}_i(t)$, so the instantaneous disagreement rate of the system is as given in (39). $\square$

**Lemma 10** (Average length of disagreement periods). *It holds that*

$$D_{avg} \leq \overline{\tau}_{\max}^{\text{sync}}. \tag{40}$$

*Proof.* To bound $D_{\text{avg}}$, it suffices to bound the expected length of a disagreement period with arbitrary initial states. Without loss of generality, consider a disagreement period on arm $i$ that starts at time $t_{\text{begin}} = 0$, with initial states $(S_i(0), \widehat{S}_i(0)) = (s, \widehat{s})$. During the disagreement period, there is no disagreement event, so as argued in the paragraph after Definition 3, the transitions of the $i$-th arm is identical to a leader-and-follower system. Therefore, we either have $S_i(t) = \widehat{S}_i(t)$ after $\tau^{\text{sync}}(s, \widehat{s})$ amount of time, or have a disagreement event before that. In either case, $t_{\text{end}} \leq \tau^{\text{sync}}(s, \widehat{s})$. Therefore, the expected length of the disagreement period satisfies

$$\mathbb{E}[t_{\text{end}} - t_{\text{begin}}] = \mathbb{E}[t_{\text{end}}] \leq \mathbb{E}[\tau^{\text{sync}}(s, \widehat{s})] \leq \overline{\tau}_{\max}^{\text{sync}}.$$

This holds for arbitrary initial states $(s, \widehat{s})$, so $D_{\text{avg}} \leq \overline{\tau}_{\max}^{\text{sync}}$. $\qquad\square$

*Proof of Lemma 7.* Combining Lemma 8, 9, and 10, we have

$$\mathbb{E}\left[\sum_{i=1}^{N} \mathbb{1}\left\{\widehat{S}_i(\infty) \neq S_i(\infty)\right\}\right] = \mathbb{E}\left[d(\boldsymbol{S}(\infty), \widehat{\boldsymbol{S}}(\infty))\right] \cdot D_{\text{avg}}$$

$$\leq 2g_{\max}\overline{\tau}_{\max}^{\text{sync}}\mathbb{E}\left[\left|\sum_{i=1}^{N} \widehat{A}_i(\infty) - \alpha N\right|\right].$$

This proves (32).

Observe that by the definition of our algorithm, at any time $t$,

$$\sum_{i=1}^{N} \mathbb{1}\left\{\widehat{A}_i(t) \neq A_i(t)\right\} = \left|\sum_{i=1}^{N} \widehat{A}_i(t) - \alpha N\right| \quad a.s.$$

Taking the steady-state expectation, we get (33). $\qquad\square$

## G   Experiment details and additional experiments

In this section, we include the details of the experiments in the main body, as well as some additional experiments. In Appendix G.1 and G.2, we describe the details of the experiments of Figure 1 and 3. We conduct the simulations of these two experiments with different initial points and present the results in Appendix G.3. To give an intuitive understanding of the difference between these policies, we also display more visualization of the sample paths under different policies in Appendix G.4. [3]

### G.1   Experiment details of Figure 1

**Restless bandits setting**   In Figure 1, we consider the discrete-time $N$-arm restless bandits represented by the tuple $(N, \mathbb{S}^N, \mathbb{A}^N, P, r, \alpha N)$. We vary the number of arms $N$, and keep the rest of the parameters fixed. The state space of each arm is $\mathbb{S} = \{1, 2, 3\}$. The action space is $\mathbb{A} = \{0, 1\}$. The transition kernel $P$ is given by

$$P(\cdot, 0, \cdot) = \begin{bmatrix} 0.02232142 & 0.10229283 & 0.87538575 \\ 0.03426605 & 0.17175704 & 0.79397691 \\ 0.52324756 & 0.45523298 & 0.02151947 \end{bmatrix},$$

$$P(\cdot, 1, \cdot) = \begin{bmatrix} 0.14874601 & 0.30435809 & 0.54689589 \\ 0.56845754 & 0.41117331 & 0.02036915 \\ 0.25265570 & 0.27310439 & 0.4742399 \end{bmatrix},$$

where the number in $s$-th row and $s'$-th column in each matrix represents $P(s, 0, s')$ or $P(s, 1, s')$, for $s, s' \in \{1, 2, 3\}$. The reward function $r$ is given by

$$r(\cdot, 0) = \begin{bmatrix} 0 & 0 & 0 \end{bmatrix},$$

---

[3]Our simulation code can be found in the link `https://github.com/YigeHong/rb-break-ugap-ftva`.

$$r(\cdot, 1) \ = \ [0.37401552 \quad 0.11740814 \quad 0.07866135],$$

where the $s$-th entry of each vector represents $r(s,0)$ or $r(s,1)$, for $s \in \{1,2,3\}$. The budget parameter $\alpha = 0.4$, so $0.4N$ arms are pulled in each time slot. This setting is taken from the Appendix E of [GGY20] as a counterexample to the UGAP assumption. We note that this RB problem obviously satisfies Synchronization Assumption for any policy: observe that $P(s, a, s') > 0$ for all $s, s' \in \mathbb{S}, a \in \mathbb{A}$, so two arms with any initial states have a positive probability of synchronizing in the next time slot, which implies finite synchronization time by Proposition 2.

**Simulation setting** We plot the long-run average reward against the number of arms for different policies. The number of arms $N$ varies from $100, 200, \ldots, 1000$. Three policies are considered: our policy FTVA($\bar{\pi}^*$), the Whittle's index policy [Whi88], and an LP-Priority policy [Ver16]. For each data point, we obtain the long-run average reward and its confidence interval by simulating 50 independent trajectories, each with a length of 1000 time slots. The initial states of all arms are simply chosen to be 1, because simulation results do not quite depend on the initial states (see Appendix G.3).

**Detail of policies** We implement Whittle index policy and LP-Priority in the standard way. The resulting priorities of both policies turn out to be $1 > 2 > 3$ (state 1 has the highest priority). This is not a coincidence given that the solution of the single-armed problem (LP) is:

$$y^*(\cdot, \cdot) \ = \ \begin{bmatrix} 0 & 0.29943 \\ 0.23768 & 0.10057 \\ 0.36232 & 0 \end{bmatrix},$$

where the $s$-th row represents $y^*(s,0)$ and $y^*(s,1)$, for $s \in \{1,2,3\}$. This solution implies that a reasonable policy should almost always pull arms in state 1, pull arms in state 2 for a certain fraction of time, and almost never pull arms in state 3. Therefore the only reasonable priority in this RB setting is $1 > 2 > 3$.

Our policy FTVA($\bar{\pi}^*$) is implemented according to the pseudocode in Section 3. The non-trivial detail here is the tie-breaking rule for selecting the set of arms to activate based on the virtual actions. Our tie-breaking rule is to select $A_i(t)$'s such that the number of good arms, i.e., the arms such that $S_i(t) = \widehat{S}_i(t)$ and $A_i(t) = \widehat{A}_i(t)$, is maximized. We have experimented with alternative tie-breaking rules, whose performance matches our theoretical results, though sometimes their asymptotic optimality requires larger values of $N$ to be observed. The relationship between various tie-breaking rules and their impact on finite-$N$ performances remains unclear in the present analysis, leaving it as a topic for future investigation.

### G.2 Experiment details of Figure 3

**Restless bandits setting** In Figure 3, we consider the discrete-time $N$-arm bandits constructed as below. Suppose the state space for each arm is $\mathbb{S} = \{0, 1, \ldots, 7\}$. Each state has a *preferred action*, which is action 1 for states $0, 1, 2, 3$, and action 0 otherwise. For an arm in state $s$, applying the preferred action moves the arm to state $(s + 1) \bmod 8$ with probability $p_{s,\mathrm{R}}$, and applying the other action moves the arm to state $(s - 1)^+$ with probability $p_{s,\mathrm{L}}$.[4] The probabilities $p_{\cdot,\mathrm{R}}$ and $p_{\cdot,\mathrm{L}}$ are given by

$$p_{\cdot,\mathrm{R}} = [0.1 \quad 0.1 \quad 0.1 \quad 0.1 \quad 0.1 \quad 0.1 \quad 0.1 \quad 0.1],$$
$$p_{\cdot,\mathrm{L}} = [1.0 \quad 1.0 \quad 0.48 \quad 0.47 \quad 0.46 \quad 0.45 \quad 0.44 \quad 0.43].$$

When the arm transitions from 7 to 0, one unit of reward is generated. Equivalently, if we consider the expected reward of applying an action at a certain state, we can define the reward function as $r(7, 0) = p_{7,\mathrm{R}}$, and $r(s, a) = 0$ for all other $s \in \mathbb{S}, a \in \mathbb{A}$. The parameter $\alpha = 1/2$, so $N/2$ arms are activated in each time slot.

**Simulation setting** We plot the long-run average reward against the number of arms for different policies. The number of arms $N$ varies from $100, 200, \ldots, 1000$. Three policies are considered: our policy FTVA($\bar{\pi}^*$), a random tie-breaking policy, and a particular LP-Priority policy that prioritizes arms with larger Lagrangian optimal indices (see the definition of Lagrangian optimal indices in

---

[4]Here the subscript L means "left", and R means "right". We are imagining the arms being lined up in a row from state 0 to state 7, and the preferred action moves an arm to the right.

[HF17, BS20, GGY22]). This particular LP-Priority policy is also referred to as the LP-Index policy in the literature. But for simplicity, we just refer to it as LP-Priority in this paper. For each data point, we obtain the long-run average reward and its confidence interval by simulating 50 independent trajectories, each with a length of 1000 time slots. The initial states for the simulations are fixed: $N/3$ arms are in state 1 and $2N/3$ arms are in state 2. Note that we fix this initial point in the simulation of Figure 3 in order to demonstrate that both the random tie-breaking and LP-Priority can get nearly zero rewards in this example. For other choices of fixed points, there is also a strong separation between the performance of FTVA($\bar{\pi}^*$) and random tie-breaking or LP-Priority, which we will show in the next section.

**Details of policies**    The optimal solution of (LP) is $y^*(s, 1) = 1/8$ for $s = 0, 1, 2, 3$, $y^*(s, 0) = 1/8$ for $s = 4, 5, 6, 7$, and $y^*(s, a) = 0$ for other $s \in \mathbb{S}$ and $a \in \mathbb{A}$. Note that the same $y^*$ remains optimal even if we remove the budget constraint. The optimal LP solution suggests that a reasonable policy should almost always pull arms in states $0, 1, 2, 3$, and almost never pull arms in states $4, 5, 6, 7$.

The random tie-breaking policy that we consider prioritizes arms whose states are in $\{0, 1, 2, 3\}$ over arms whose states are in $\{4, 5, 6, 7\}$, and it breaks ties uniformly at random when there are more than $N/2$ arms in either of the two sets.

The LP-Priority policy we consider prioritizes arms with larger Lagrangian optimal indices [BS20, GGY22]. Specifically, it involves solving the Lagrangian relaxation of the original LP, which replaces the budget constraint with a penalty term determined by the optimal Lagrange multiplier. In our setting, because the optimal solution $y^*$ remains optimal even without the budget constraint, we can simply remove the budget constraint to get the Lagrangian relaxation. [5] The resulting Lagrangian optimal indices are given by:

$$[0.0125 \quad 0.1375 \quad 0.0725 \quad 0.07125 \quad -0.07 - 0.06875 \quad -0.0675 \quad -0.06625],$$

and the priority is $1 > 2 > 3 > 0 > 7 > 6 > 5 > 4$.

The implementation of our policy FTVA($\bar{\pi}^*$) is the same as in the last experiment.

**Further discussions of the policies**    As shown in Figure 3, the random tie-breaking policy and the LP-Priority policy get nearly zero rewards. We have discussed why this happens for the random tie-breaking policy in Section 3.3. For the LP-Priority policy based on the Lagrangian optimal index, the reason why it does not work is less obvious. Some sample paths suggest that under this policy, the arms will concentrate on $\{3, 4, 5\}$ most of the time and thus cannot get a reward. Here is a possible explanation for why arms cannot easily escape from $\{3, 4, 5\}$: when some of the arms transition to state 6, by the priority $6 > 5 > 4$, those arms in state 6 are likely to be activated. Because states 6 do not prefer action 1, those arms will transition back to state 5 and thus get trapped.

In addition to the numerical results, we also note that our policy FTVA($\bar{\pi}^*$) is provably asymptotically optimal in this RB problem even though its transition kernel is not unichain. This stems from the fact that the optimal single-armed policy $\bar{\pi}^*$ has a single recurrent class $\mathbb{S}$, and satisfies Synchronization Assumption. See Appendix D for the discussion on the conditions under which Synchronization Assumption and Theorem 1 hold when unichain is not assumed.

### G.3    Varying the initial points of the Figure 1 and 3 examples

In Figure 1 and Figure 3, we have fixed one initial point for each simulation, for the ease of presentation. In this section, we rerun the simulations for these two settings with more initial points. For the simulations of each setting, we generate initial states in the following way.

- We first choose a probability distribution on the state space $\mathbb{S}$ from all the possible probability distributions on $\mathbb{S}$ uniformly at random. Let $\pi(s)$ denote the probability of state s under the chosen distribution. This distribution is chosen independently across the 20 sets of simulations.
- For each $N$-armed problem, we set the initial state such that $N \cdot \pi(s)$ arms are in state s for each s, with proper rounding.
- For each $N$-armed problem, we run all the policies with this initial state.

---

[5]A nuance is that the optimal Lagrange multiplier for the budget constraint is not unique in this setting, so there can be different Lagrange relaxations. We focus on the simplest one.

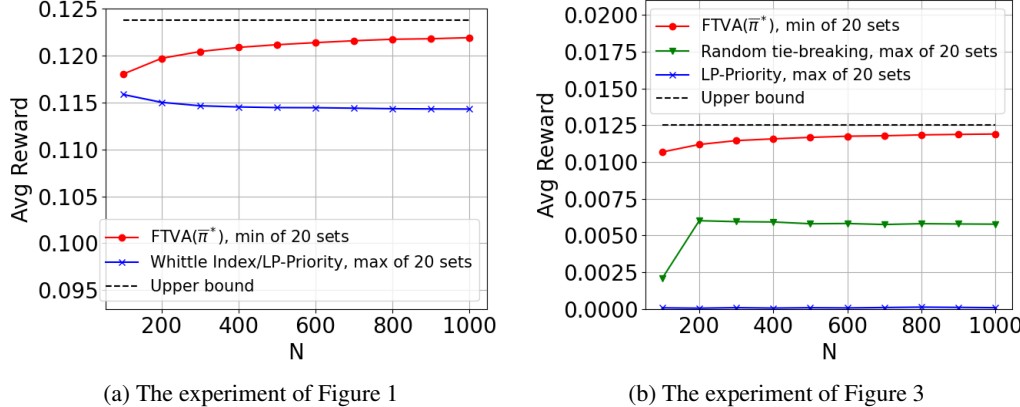

(a) The experiment of Figure 1        (b) The experiment of Figure 3

Figure 6: We re-run the two experiments of Figure 1 and 3 with 20 sets of simulation with randomly generated initial points. For FTVA, we calculate the *minimum* average reward over the 20 sets; for the other policies, we calculate the *maximum* average reward over the 20 sets.

We illustrate the results of the 20 sets of simulations in the Figure 6a and 6b in the following way: For FTVA($\bar{\pi}^*$), for each $N$, we take the minimum average reward over the 20 sets of simulations and plot them in the figure; For other policies, for each $N$, we take the maximum average reward over the 20 sets of simulations. We can see that the min curve for FTVA($\bar{\pi}^*$) still approaches the optimal value as $N$ increases, whereas the max curves for other policies are strictly separated from the optimal value.

### G.4 Visualization of the policies

In Figure 2 of Section 3.3, we illustrate what LP-Priority and FTVA($\bar{\pi}^*$) does differently by visualizing their sample paths. Here we include more such figures.

We still use the same example defined in Section 3.3. To recap, this example has 8 states, numbered as $\{0, 1, 2, 3, 4, 5, 6, 7\}$. The optimal single-armed policy always chooses the preferred action in each state, which is 1 for states $0, 1, 2, 3$, and 0 for states $4, 5, 6, 7$. Under the optimal single-armed policy, each arm moves from state $s$ to $(s + 1) \mod 8$ with probability $0.1$ or stays at $s$ with probability $0.9$. The optimal distribution for the single-armed system is uniform over the 8 states.

When simulating the policies in the $N$-armed system, we can tell that a policy is good if the fraction of arms in each state is roughly uniform, and bad if the arms concentrate on a small number of states. We can also see why this happens by looking at whether the preferred action is chosen, and whether the arms are moving from $s$ to $(s + 1) \mod 8$.

The three heatmaps in Figure 7a illustrate how the fractions of arms in each state change over time under the three policies: random tie-breaking, LP-Priority, and FTVA($\bar{\pi}^*$). The x-axis represents the time slot, which ranges from 0 to 499; the y-axis represents the states; the brighter color represents a larger fraction of arms in this state at this time, and the specific value represented by each color can be found in the color bar on the right.

We can see that under the random tie-breaking, the arms concentrate around state 0; under LP-Priority, the arms concentrate around states $\{3, 4, 5\}$ after a burn-in period; under FTVA($\bar{\pi}^*$), the arms uniformly spread out over the 8 states.

In Figure 7b, we take a closer look at the sample path of the random tie-breaking policy from time 250 to 289 and contrast it with the sample path if the system uses FTVA($\bar{\pi}^*$) from time 250 onwards. We also add some arrows indicating the *drift* of the arms, i.e., the average direction that the arms in this state are moving into. The colors of the arrows represent the direction: the blue arrow implies that the arm takes the preferred action and moves in the correct direction, while the red arm implies that the arm takes the non-preferred action and moves in the wrong direction. The magnitudes of the arrows represent how fast they are moving.

We can see that under random tie-breaking, although the arms have a strong tendency to move up from state 0 to state 1, they move back when they reach state 1 and thus get stuck at state 0. In

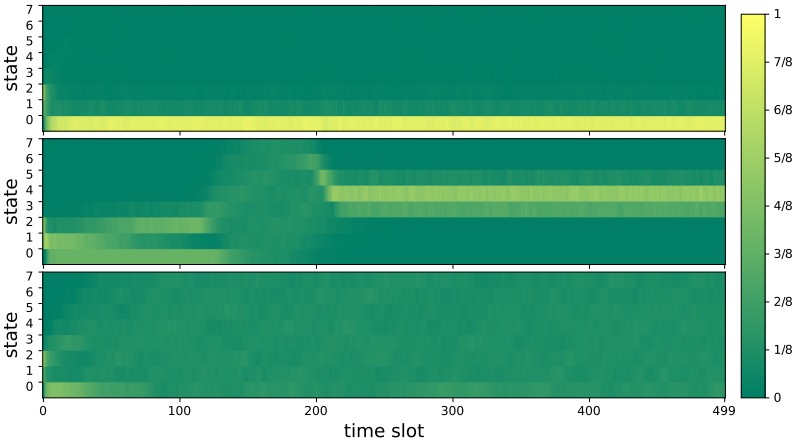

(a) Time evolution of the fractions of arms in each state under the three policies during the first $500$ time slots. Top to bottom: Random Tie-Breaking, LP-Priority, and $\texttt{FTVA}(\bar{\pi}^*)$.

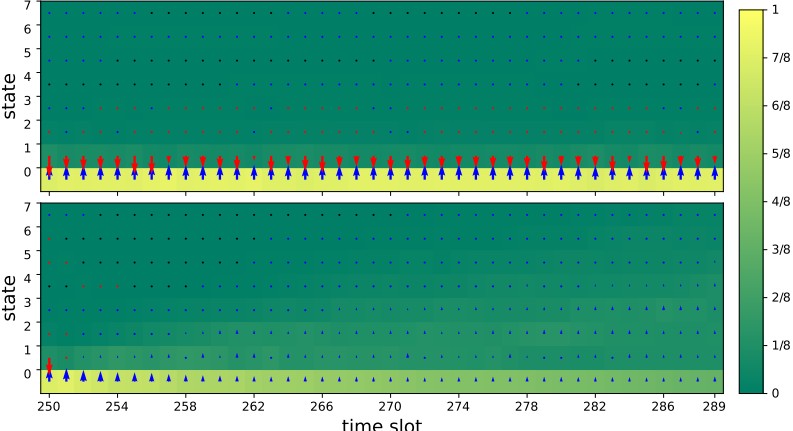

(b) Time evolution of the fraction of arms in each state under Random Tie-Breaking (upper), or after switching to $\texttt{FTVA}(\bar{\pi}^*)$ (lower) since time slot $250$. The color and magnitude of the arrows represent the average movement of the arms in each state.

Figure 7: Visualization of the sample paths under different policies for the example in Section 3.3.

contrast, when switching to $\texttt{FTVA}(\bar{\pi}^*)$, most arrows point in the right direction, which implies that the arms consistently apply the preferred actions. As a result, $\texttt{FTVA}(\bar{\pi}^*)$ helps the arms to escape from state $0$ and converge to the uniform distribution over the state space.

Figure 8a illustrates the fraction of arms in state $s$ taking action $a$ in a few time slots after $250$ in the same sample path of the random tie-breaking policy in Figure 7a. For each $s$ and $a$. The upper bar plot is under the random tie-breaking policy, while the lower bar plot is after switching to $\texttt{FTVA}(\bar{\pi}^*)$. The x-axis represents the state; the y-axis represents the fraction; there are four bars in each state, corresponding to $4$ time-steps. Each bar has two segments, where the length of the blue segment indicates the fraction of arms taking the preferred action in this state, and the length of the red segment indicates the fraction of arms taking the non-preferred action in this state; the lower segment of the bar corresponds to action $0$, and the higher segment of the bar corresponds to action $1$.

We can see that under $\texttt{FTVA}(\bar{\pi}^*)$, more arms choose the preferred actions for state $1$ than under the random tie-breaking policy, which prevents the arms in state $1$ from moving back to state $0$ as we see in Figure 7b under the random tie-breaking policy.

Figure 8b is similar to Figures 8a, except that the random tie-breaking policy is replaced by the LP-Priority policy. We can again see that under LP-Priority there is a force that causes the arms to concentrate on a state, whereas switching to $\texttt{FTVA}(\bar{\pi}^*)$ helps the system to escape from that state.

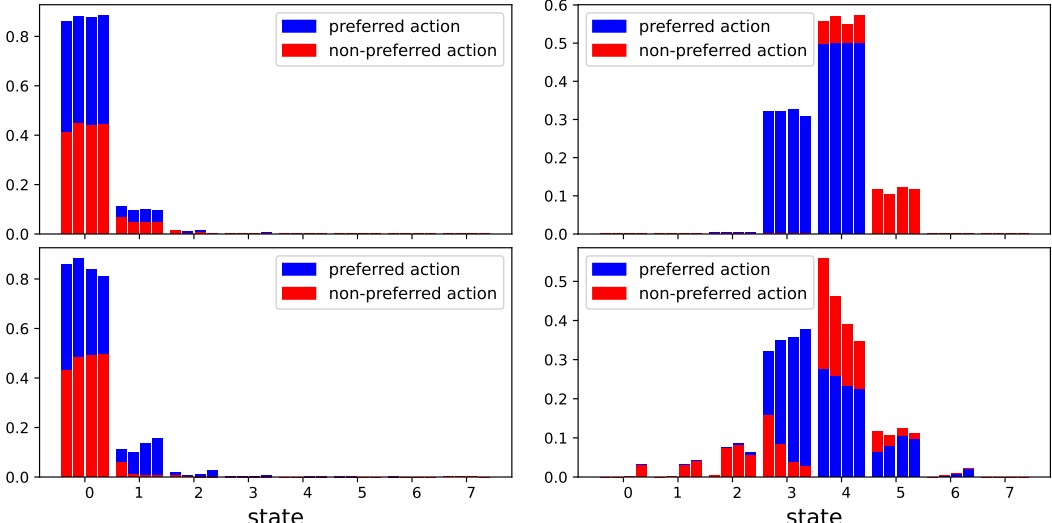

(a) The fraction of arms in each state taking each action in the 4 time steps following time 250, under Random Tie-Breaking (upper plot) or after switching to FTVA($\bar{\pi}^*$) (lower plot).

(b) The fraction of arms in each state taking each action in the 4 time steps following time 250, under LP-Priority (upper plot) or after switching to FTVA($\bar{\pi}^*$) (lower plot).

Figure 8: Actions taken by different policies for the example in Section 3.3. In each bar plot, the X-axis represents the states; the Y-axis represents the fractions. Each state has bars for the 4 time steps. The blue segment of each bar is the fraction of arms taking the preferred action; the red segment corresponds to the non-preferred action. The lower segment of each bar corresponds to action 0; the higher segment corresponds to action 1.

The common failure mode for random tie-breaking and LP-Priority in this setting is that the arms concentrate on a bad state $s$, which prevents the arms on state $(s + 1)$ from applying the preferred action due to the budget constraint. This causes a *livelock* where the arms move back and forth between state $s$ and $(s + 1)$, and fail to gain a reward by moving past state 7. FTVA($\bar{\pi}^*$) solves this issue by letting more arms in state $(s + 1)$ follow the preferred action, which breaks the livelock and helps all arms converge to the optimal distribution.

# H    Generalization to heterogeneous arms

In this section, we show how FTVA and its analysis can be extended to the case with heterogenous arms. We focus on the discrete-time case for simplicity. The continuous-time case can be analyzed in a similar fashion.

## H.1    Setting and algorithm

Suppose the arms are divided into $K$ types, with $\beta_k N$ arms in each type $k \in \{1, 2, \ldots, K\} \triangleq [K]$, and each type is associated with an MDP. The transition kernels and reward functions can be different across types. Suppose each arm is indexed by $i \in [N]$, and we denote the type of the $i$-th arm as $k(i)$. For any type $k$ arm, we let $P_k(s, a, s')$ be its probability of transitioning from state $s$ to $s'$ upon taking action $a$, and let $r_k(s, a)$ be its reward for taking action $a$ in state $s$.

Consider the linear program below, whose variables $y_k(s, a)$ represents the steady-state probability that a type $k$ arm is in state $s$ taking action $a$, for $k \in [K], s \in \mathbb{S}, a \in \mathbb{A}$.

$$\underset{\{y_k(s,a)\}_{k \in [K], s \in \mathbb{S}, a \in \mathbb{A}}}{\text{maximize}} \quad \sum_k \sum_{s,a} \beta_k r_k(s, a) y_k(s, a) \tag{LP-Het}$$

$$\text{subject to} \quad \sum_k \sum_s \beta_k y_k(s, 1) = \alpha, \tag{41}$$

$$\sum_{s',a} y_k(s',a) P_k(s',a,s) = \sum_a y_k(s,a), \quad \forall k \in [K], s \in \mathbb{S}, \qquad (42)$$

$$\sum_{s,a} y_k(s,a) = 1 \ \ \forall k \in [K]; \qquad (43)$$

$$y_k(s,a) \geq 0, \ \ \forall k \in [K], s \in \mathbb{S}, a \in \mathbb{A}.$$

where the three constraints (41), (42), and (43) correspond to (5), (6), and (7) in (LP); when writing summations, we omit $\in [K], \in \mathbb{S}$ and $\in \mathbb{A}$ for simplicity.

(LP-Het) can be viewed as a relaxation of the $N$-armed problem. To see this, take any $N$-armed policy $\pi$ and set $y_k(s,a)$ to be the fraction of arms in state $s$ taking action $a$ among type $k$ arms in steady state under $\pi$, i.e.,

$$y_k(s,a) = \frac{1}{\beta_k N} \mathbb{E}\Big[ \sum_{k(i)=k} \mathbb{1}_{\{S_i^\pi(\infty)=s, A_i^\pi(\infty)=a\}} \Big].$$

Whevener $\pi$ satisfies the budget constraint (2), $\{y_k(s,a)\}_{k\in[K],s\in\mathbb{S},a\in\mathbb{A}}$ satisfies (41)–(43). Therefore, the optimal value of (LP-Het), which we denote as $V_1^{\text{rel}}$, is an upper bound of the optimal value of the $N$-armed problem, i.e., $V_1^{\text{rel}} \geq V_N^*$.

The optimal solution to (LP-Het), $\{y_k^*(s,a)\}_{k\in[K],s\in\mathbb{S},a\in\mathbb{A}}$, induces a optimal single-armed policy $\bar{\pi}_k^*$ for each type $k \in [K]$:

$$\bar{\pi}_k^*(a|s) = \begin{cases} y_k^*(s,a)/(y_k^*(s,0)+y_k^*(s,1)), & \text{if } y_k^*(s,0)+y_k^*(s,1) > 0, \\ 1/2, & \text{if } y_k^*(s,0)+y_k^*(s,1) = 0. \end{cases} \quad \text{for } s \in \mathbb{S}, a \in \mathbb{A}. \qquad (44)$$

Standard argument in [Put05] show that if a type $k$ arm runs $\bar{\pi}_k^*$, it achieves the steady-state expected reward $\sum_{s,a} r_k(s,a) y_k^*(s,a)$, and requires $\sum_s y_k^*(s,1)$ unit of budget in steady-state. If each type $k$ arm could independently run the optimal single-armed policy of its type, since the fraction of type $k$ arms in the $N$-armed system is $\beta_k$, in steady state, the expected reward per arm would be $V_1^{\text{rel}}$, and the budget requirement per arm would be $\sum_{k=1}^K \sum_s y_k^*(s,1) = \alpha$, which is analogous to the homogeneous case where the optimal policy of the relaxed problem in (3) achieves $V_1^{\text{rel}}$ reward and requires $\alpha$ expected budget in steady-state.

To convert from the single-armed policies to an $N$-armed policy, FTVA lets each arm of type $k$ independently simulate a virtual single-armed process following $\bar{\pi}_k^*$. Lines 3-14 of Algorithm 1 stay the same. We denote the resulting policy as $\texttt{FTVA}(\{\bar{\pi}_k^*\}_{k\in[K]})$. The resulting policy can be proved to achieve $O(1/\sqrt{N})$ optimality gap, under the assumption that Synchronization Assumption is satisfied for each $\bar{\pi}_k^*$, as stated below:

**Theorem 3.** *In restless bandits with heterogeneous arms, let $\{\bar{\pi}_k^*\}_{k\in[K]}$ be the optimal single-armed policies induced by* (LP-Het). *Assume that for each $k \in [K]$, $\bar{\pi}_k^*$ satisfies Synchronization Assumption with the synchronization times $\{\tau_k^{\text{sync}}(s,a,\widehat{s},\widehat{s})\}_{(s,a,\widehat{a})\in\mathbb{S}\times\mathbb{A}\times\mathbb{S}\times\mathbb{A}}$. For any $N \geq 1$, the optimality gap of $\texttt{FTVA}(\{\bar{\pi}_k^*\}_{k\in[K]})$ is upper bounded as*

$$V_N^* - V_N^{\texttt{FTVA}(\{\bar{\pi}_k^*\}_{k\in[K]})} \leq \frac{r_{\max} \overline{\tau}_{\max}^{\text{sync}}}{\sqrt{N}}, \qquad (45)$$

*where $r_{\max} \triangleq \max_{s\in\mathbb{S},a\in\mathbb{A}} |r(s,a)|$ and $\overline{\tau}_{\max}^{\text{sync}} \triangleq \max_{k\in[K],(s,a,\widehat{s},\widehat{a})\in\mathbb{S}\times\mathbb{A}\times\mathbb{S}\times\mathbb{A}} \mathbb{E}\big[\tau_k^{\text{sync}}(s,a,\widehat{s},\widehat{a})\big].$*

### H.2 Proof for Theorem 3

In this section, we prove Theorem 3.

Since $V_1^{\text{rel}} \geq V_N^*$, it suffices to show that

$$V_1^{\text{rel}} - V_N^{\texttt{FTVA}(\{\bar{\pi}_k^*\}_{k\in[K]})} \leq \frac{r_{\max} \overline{\tau}_{\max}^{\text{sync}}}{\sqrt{N}}. \qquad (46)$$

We start with an inequality that has the same form as (13) in the homogeneous case:

$$V_1^{\text{rel}} - V_N^{\texttt{FTVA}(\{\bar{\pi}_k^*\}_{k\in[K]})} = \frac{1}{N} \mathbb{E}\left[ \sum_{i=1}^N r\big(\widehat{S}_i(\infty), \widehat{A}_i(\infty)\big) - \sum_{i=1}^N r\big(S_i(\infty), A_i(\infty)\big) \right]$$

$$\leq \frac{2r_{\max}}{N} \mathbb{E}\left[\sum_{i=1}^{N} \mathbb{1}\left\{\left(\widehat{S}_i(\infty), \widehat{A}_i(\infty)\right) \neq \left(S_i(\infty), A_i(\infty)\right)\right\}\right], \quad (47)$$

where in the first equality, we used the fact that the virtual processes are independently running the single-armed policy of the corresponding types, so they achieve the average reward $V_1^{\text{rel}}$.

Defining the disagreement events, disagreement periods, and disagreement rates in the same way as Appendix E and applying Little's law, we have

$$\mathbb{E}\left[\sum_{i=1}^{N} \mathbb{1}\left\{\left(\widehat{S}_i(\infty), \widehat{A}_i(\infty)\right) \neq (S_i(\infty), A_i(\infty))\right\}\right] = \mathbb{E}\left[d(\widehat{\boldsymbol{S}}(\infty))\right] \cdot D_{\text{avg}}, \quad (48)$$

where $d(\widehat{\boldsymbol{s}})$ is the instantaneous disagreement rate, and $D_{\text{avg}}$ is the long-run average length of disagreement periods. With (47) and (48), it remains to bound $\mathbb{E}\left[d(\widehat{\boldsymbol{S}}(\infty))\right]$ and $D_{\text{avg}}$.

By the definition of the policy, we have $\mathbb{E}\left[d(\widehat{\boldsymbol{S}}(\infty))\right] = \mathbb{E}\left[\left|\sum_{i=1}^{N} \widehat{A}_i(\infty) - \alpha N\right|\right]$, $\widehat{A}_i(\infty)$'s are independent Bernoulli random variables, and $\mathbb{E}\left[\widehat{A}_i(\infty)\right] = y_k^*(s, 1)$ if arm $i$ is of type $k$. As a result,

$$\mathbb{E}\left[\sum_{i=1}^{N} \widehat{A}_i(\infty)\right] = \sum_k \beta_k N \sum_{s,a} y_k^*(s, 1) = \alpha N.$$

Using the same Cauchy-Schwartz argument as in the proof of the homogeneous case in Appendix E, it is not hard to show that $\mathbb{E}\left[d(\widehat{\boldsymbol{S}}(\infty))\right] \leq \sqrt{N}/2$.

As for $D_{\text{avg}}$, it is not hard to see from the definition that the length of each disagreement period of a type $k$ arm is stochastically dominated by $\tau_k^{\text{sync}}(s, a, \widehat{s}, \widehat{a})$, where $(s, a, \widehat{s}, \widehat{a})$ are the initial states and actions of that disagreement period. Therefore, the average length of disagreement period is bounded by

$$D_{\text{avg}} \leq \max_{k \in [K], (s,a,\widehat{s},\widehat{a}) \in \mathbb{S} \times \mathbb{A} \times \mathbb{S} \times \mathbb{A}} \mathbb{E}\left[\tau_k^{\text{sync}}(s, a, \widehat{s}, \widehat{a})\right] = \overline{\tau}_{\max}^{\text{sync}}.$$

Combining the above calculations, we get (46), which finishes the proof.

