# OpenReview forum: "Restless Bandits with Average Reward: Breaking the Uniform Global Attractor Assumption"
_NeurIPS.cc/2023/Conference — NeurIPS 2023 spotlight_

### Official Review · Reviewer_TTwr · 2023-07-03

**Soundness:** 2 fair
**Presentation:** 2 fair
**Contribution:** 2 fair
**Rating:** 4
**Confidence:** 3

**Summary:**

This paper studies the infinite-horizon restless bandits problem and proposed a simulation-based framework, i.e., Follow-the-Virtual-Advice, to leverage single armed policy to solve a multi-armed problem, which gets rid of the difficult-to-verify condition, i.e., the uniform global attractor property. Removing this pre-condition is a significant improvement for RMABs.

**Strengths:**

This paper studies the infinite-horizon restless bandits problem and proposed a simulation-based framework, i.e., Follow-the-Virtual-Advice, to leverage single armed policy to solve a multi-armed problem, which gets rid of the difficult-to-verify condition, i.e., the uniform global attractor property. Removing this pre-condition is a significant improvement for RMABs.

**Weaknesses:**

1. The arguments on FTVA in Section 3.3 look to be a bit hands-waving since there is no clear evidence to support their claims. Based on the reviewer’s understandings, those arguments are possibly wrong.  For example, in lines 181-184, the authors claim that even if the initial virtual state and real state of an arm are different, they will become identical in finite time by chance under mild assumptions in Section 4.1. However, Assumption 1 relies on the “observation” in lines 233-238, which is not true. All the transitions are stochastic, not deterministic. How could we guarantee that $S(t+1) =\hat{S}(t+1)$ if $S(t)$ is different with $\hat{S}(t)$?  If Assumption 1 fails, the argument in lines 183-184 does not hold. The arguments in lines 185-189 are also not true. Even when the virtual process can always satisfy the budget constraint, how do we guarantee that the real process following the virtual actions does not violate the budget constraints.

2. The example in lines 194-215 is also not fully supported by evidence. The single-agent policy defined in eq. (8) is stationary stochastic policy, and agent selects actions according to certain probabilities. What do you mean by the preferred action at each state? Though this particular example may only have one action at each state, there is no preferred action in general. How to guarantee the descriptions in lines 207-209 to be true?  This example shows that the real and virtual process has different state distributions, which contradicts Section 3.3. See comment 1.

3. A very important related work is missing. [Ghosh22] also gets rid of the global attractor assumption and considers a much more challenging setting with heterogeneous arms and multi-actions.  An outstanding limitation of this work is that all arms must be the same.

Ghosh, A., Nagaraj, D., Jain, M., & Tambe, M. (2022). Indexability is Not Enough for Whittle: Improved, Near-Optimal Algorithms for Restless Bandits. arXiv preprint arXiv:2211.00112.

4. The references cited in this paper are not precise. For example, as far as the reviewer knows, [Ver16] does not consider the single-armed problem. Hence, the argument in lines 140-141 are not precise.  Another example is in lines 123-124. The reviewer is not aware of the meaning of these sentences. How are they related to the RMAB problem in eqs. (1)-(2)?


**Questions:**

See comments in Weaknesses.

**Limitations:**

no negative societal impact.

---

> ### Author Rebuttal · Authors · 2023-08-09
>
> We appreciate the reviewer’s time and effort put into the review. The reviewer has two main concerns: correctness of our results (Comments 1 and 2), and relationship with prior work by Ghosh et al. (ArXiv 22, AAMAS 23) (Comment 3). Regarding the correctness of our results, we believe that our proofs are rigorous, as explained in detail below. We look forward to further conversation with the reviewer, so that we can have an opportunity to resolve any remained confusion and answer additional questions. Moreover, we are open to suggestions that may improve the readability of our arguments. Regarding the relation with [Ghosh23] and a related comment on the homogeneous arms in our paper, we respond to these points in the **global rebuttal**.
>
> > Comment 1: Arguments on FTVA in Section 3.3.
>
> Firstly, we’d like to clarify that Section 3.3 is meant to provide insights and intuitions for FTVA. Rigorous proofs are provided in appendices. Building on that, we strive for making the presentation as informative and helpful as possible. Below we respond to specific concerns from the reviewer.
>
> - > Lines 181-184 and Assumption 1 in lines 233-238
>
>     We’d like to first clarify that lines 233-238 describe how we *define* the leader-and-follower system, rather than representing an observation. Next we’d like to better explain this system. As the reviewer noted, when $S(t) \neq \widehat{S}(t)$, it is possible that $S(t+1)\neq\widehat{S}(t+1)$ since they’re sampled independently and there’s randomness in the transitions. However, Assumption 1 does not require the real state and virtual state of an arm to become identical in just a single time step. The time duration needed for them to become identical again is random, and Assumption 1 only requires this random time duration to have a finite expectation. To make Assumption 1 easier to grasp, we have provided several sufficient conditions in Appendix A that are more intuitive and easier to verify.
>
> - > Even when the virtual process can always satisfy the budget constraint, how do we guarantee that the real process following the virtual actions does not violate the budget constraints.
>
>     By the design of FTVA (Algorithm 1), when the virtual actions satisfy the budget constraint at a time step t, the real actions will be identical to the virtual actions, thus satisfying the budget constraint as well. Even when the virtual actions do not satisfy the budget constraint, the real actions still always satisfy the budget constraint. This is guaranteed by lines 3-7 in the pseudocode of Algorithm 1. For the real actions, we apply action 1 to arms in the set $\mathcal{A}$, which is a set of cardinality $\alpha N$ (the budget).
>
> > Comment 2: Example in lines 194-215.
> - > Preferred action
>
>     We apologize for the potential confusion caused by our wording "preferred action". In this example, each state has two possible actions, 1 and 0 (as in any restless bandit problem). We call action 1 the "preferred action" for states 0, 1, 2, 3; we call action 0 the "preferred action" for states 4, 5, 6, 7. We chose this naming since in this example, the optimal single-armed policy is to apply action 1 for states 0, 1, 2, 3, and apply action 0 for states 4, 5, 6, 7. (In the N-armed system, some arms may not be able to actually apply their preferred actions due to the budget constraint.) We will make sure to clarify this in the revision.
>
> - > Lines 207-209
>
>     Lines 207-209 follow from the optimal solution of the LP in (5)-(7) and the induced optimal policy in (8). Under this optimal policy, one can verify that the state distribution in steady-state is uniform on $\mathbb{S}$. Also note that the virtual processes are independent. Therefore, by law of large numbers, roughly N/2 arms have virtual states in the set $\{4,5,6,7\}$.
>
> - > State distributions of real and virtual processes
>
>     Although the initial state distributions of the real and virtual processes are different, the state distributions will become the same in *steady state,* where the steady-state probability of being in each state is equal to the long-run fraction of time spent in that state.
>
> > Comment 3: Relation with [Ghosh23] and assumption of homogeneous arms.
>
> We thank the reviewer for bringing up this related work and the homogeneity assumption. We address these two important points in detail in our **global rebuttal**. For immediate reference, we provide a brief summary here: (1) Our work differs from [Ghosh23] in terms of problem setting (finite-horizon or discounted-reward in [Ghosh23], infinite-horizon average-reward in our paper) and algorithmic approach; moreover, significant challenges arise when one attempts to apply techniques from [Ghosh23] to our setting. (2) Our paper focuses on breaking the UGAP assumption for the infinite-horizon average-reward setting, which is challenging even with homogeneous arms. That said, our approach and results can be directly generalized to accommodate heterogeneous arms in a wide range of settings.
>
> > Comment 4: Single-armed problem in lines 140-141.
>
> Although some prior work may not use the term "single-armed problem", *equivalent* forms of this problem are used. In [Ver16], the LP in (4)-(7) can be viewed as a single-armed problem since there is a standard way to construct an optimal policy from the solution (see, e.g., Chapters 8.8 & 8.9 in [Put05]). The construction process is similar to our equation (8). We will make this connection between the LP and the single-armed problem more explicit in our revision.
>
> > Comment 4: Lines 123-124.
>
> The RMAB problem in equations (1)-(2) is a finite-state MDP. As stated in lines 123-124, it is generally accepted that for finite-state MDPs, it suffices to consider Markovian stationary policies. This result is widely referenced; a proof can be found in, e.g., Chapter 5.5 of [Put05]. We included this point in our paper to make it clear that our approach deviates from some traditional approaches.

---

> > ### Comment · Area_Chair_qMx2 · 2023-08-16
> > **Contrasts between discounted and average reward setting**
> >
> > The authors have claimed that the related work the reviewer mentions is for a different technical setting. I can corroborate this statement. Therefore, can the reviewer please consider the contribution of this work in light of these developments?

---

### Official Review · Reviewer_S8QK · 2023-07-05

**Soundness:** 3 good
**Presentation:** 3 good
**Contribution:** 4 excellent
**Rating:** 7
**Confidence:** 4

**Summary:**

This paper presents an algorithm providing an asymptotically optimal policy to solve restless bandits with N arms when N goes to infinity without using classical assumptions on the expected deterministic dynamics (uniform global attractor).


**Strengths:**



1. The idea to couple the actual trajectory under the relaxed policy with a virtual one, sampled independently for each bandit is a very nice idea.

2. Removing the UGAP assumption is also a strong point, although this is not the only paper going in this direction (see for example Zhang X, Frazier PI (2022) Near-optimality for infinite-horizon restless bandits with many arms. arXiv preprint arXiv:2203.15853).


3. The paper makes a good literature review and a good  use of previous work in this domain.

**Weaknesses:**

1. The discussion on the synchronization assumption, and especially the sufficient conditions could be made more precise. The link with UGAP is also missing.
Also there is a gap between the synchronization of each arm with the synchronization of the whole bandit (see for example the PhD thesis of Kimang Khun).

2. The fact the algorithm is based on independent sampling of each arm makes the \sqrt{N} bound tight, wuth little hope for improvement, while other approaches based on the fluid dynamics could still work with UGAP (using Cezaro averaging for example) with a higher chance to have a better convergence rate.

3. The comparison with previous algorithms is a little unfair, especially because of initial conditions.

**Questions:**

1. Can you change the initial conditions of the numerical experiments to make a fairer comparisons with existing solutions.

2. Provide answers to the weak points listed above.

**Limitations:**

I did not see any limitations.

---

> ### Author Rebuttal · Authors · 2023-08-09
>
> We thank the reviewer for recognizing the strengths and contribution of our paper. Below we provide our responses to the comments.
>
> > Discussion on the synchronization assumption and its link with UGAP
>
> In our rebuttal to Reviewer JXov, we give a detailed discussion on SA and UGAP. We repeat it here for ease of reference. This contrast is in the discrete-time, infinite-horizon, average-reward setting, given that we remove UGAP without assuming SA in the continuous-time setting. We’re open to more concrete feedback on improving the discussion.
>
> 1. **There exist problem instances where SA holds but UGAP doesn’t.**
> One limitation of our work is indeed that it is unclear whether SA subsumes UGAP (recognized in lines 61-62). However, there exist problem instances where SA holds but UGAP doesn’t. Two such examples are given in our paper (illustrated in Figures 1 and 2), where the first example is borrowed from [GGY20]. There are more examples provided in [GGY20] that do not satisfy UGAP. One can easily see that these examples satisfy SA: all entries in the two transition matrices are positive, so two arms starting from any pair of initial states have a positive probability of synchronizing in one step (a formal argument is given in Proposition 2 in Appendix A). For these examples, FTVA achieves an $O(1/\sqrt{N})$ optimality gap, whereas previously, no policy was known to be asymptotically optimal. Another perspective to view this comparison is that SA uncovers the feasibility of achieving asymptotic optimality for more problems.
>
>     Another possibly helpful comparison is the following. For any problem instance, we can make it satisfy SA by perturbing entries of its transition kernels by an arbitrarily small amount such that all the entries are strictly positive. This implies that the set of problem instances that satisfy SA is dense in the problem space. In contrast, UGAP assumes the existence of a global attractor. It can be shown that the set of instances that do not satisfy UGAP contains open sets.
>
> 2. **SA is easier to verify.**
>
>     It has been acknowledged in many prior papers that UGAP is tricky to verify. It lacks intuitive sufficient conditions. UGAP has been assessed through numerical experiments where one simulates the mean-field ODE with randomly generated initial states. But since not all initial states are can be covered, this method can confirm *violations* of UGAP rather than compliances.
>
>     In contrast, SA can be efficiently verified in several ways. By definition, SA is a hitting time for the finite-state Markov chain $(S(t), \widehat{S}(t))$ (eq. (10)) to reach the subset of states $\lbrace (s,\hat{s})\colon s=\hat{s} \rbrace$. Therefore, SA can be efficiently verified using path-finding graph algorithms on the state transition diagram of the Markov chain. Additionally, we provide several intuitive sufficient conditions for SA in Appendix A, which can also be efficiently verified by identifying self-loops and cycles.
>
> > Synchronization of each arm vs synchronization of the whole bandit
>
> We unfortunately didn’t find any reference to this distinction in the PhD thesis of Kimang Khun. We would like to invite the reviewer to further elaborate on this point so we can have an opportunity to respond further. Our synchronization assumption is made in the context of a single arm.
>
> > Hope for improving upon the $\sqrt{N}$ convergence rate
>
> We've also been contemplating this very point raised by the reviewer. Indeed, the independent sampling is subject to an error of $\Theta(\sqrt{N})$ due to the central limit theorem. But it provides a way to break UGAP. A fluid-based policy can achieve an exponential convergence rate, but it relies on UGAP to converge to a neighborhood of the optimal point. It might be possible to design a hybrid policy that uses a sampling-based policy for its global convergence benefit, and switches to a fluid-based policy around the optimal point for a faster convergence rate.
>
> > Better comparison with previous algorithms
>
> We thank the reviewer for raising this point. We have rerun the experiments demonstrated in Figure 1 and Figure 2 in our paper, and we illustrate the results in Figure R1(a) and Figure R1(b) in the pdf file attached to the global rebuttal. In each experiment, we perform 20 sets of simulations. For each set of simulation, we generate initial states in the following way.
>
> - We first choose a probability distribution on the state space $\mathbb{S}$ from all the possible probability distributions on $\mathbb{S}$ uniformly at random. Let $\pi(s)$ denote the probability of state s under the chosen distribution. This distribution is chosen independently across the 20 sets of simulations.
>
> - For each N-armed bandit, we set the initial state such that $N\cdot\pi(s)$ arms are in state s for each s, with a proper rounding.
>
> - For each N-armed bandit, we run all the policies with this initial state.
>
> We illustrate the results of the 20 sets of simulations in the figure in the following way. For the FTVA policy, for each N, we take the minimum average reward over the 20 sets of simulations, and plot them in the figure. For other policies, for each N, we take the maximum average reward over the 20 sets of simulations. We can see that the min curve for FTVA still approaches the optimal value as N increases, whereas the max curves for other policies are strictly separated from the optimal value.

---

### Official Review · Reviewer_s6TA · 2023-07-05

**Soundness:** 3 good
**Presentation:** 3 good
**Contribution:** 4 excellent
**Rating:** 6
**Confidence:** 3

**Summary:**

In the late 80s and early 90s, Whittle formalized the restless multi-armed bandit framework and Weber & Weiss proved that Whittle’s index policy, which solves a relaxed version of the optimization problem, is asymptotically optimal under conditions of indexability. Since then, RMAB problems have attracted much attention for their wide application in diverse areas. This paper contributes a new technique that is not an index policy. Instead, the simulation framework requires a new “synchronization assumption” for the discretized time case. The contributions of this paper are purely theoretical, including a new algorithmic approach (Follow-the-Virtual-Advice) and optimality gap bounds.

**Strengths:**

This paper introduces an original technique for solving two-action RMAB problems in both discretized and continuous time spaces, and provides bounds on conversion loss and gaps in optimality for both. It is clear that the only assumption that must be enforced in the discrete case is the new “synchronization assumption”. There is a nice discussion about this assumption and when it holds in the appendix for novice readers. The submission appears to be technically sound, and claims are well supported by theoretical analysis.


**Weaknesses:**

After decades of essentially little-to-no change in the state-of-the-art techniques that are deployed in practice, it is wonderful to see new approaches. The authors should be made aware of [1], which aims to address the same problem and also does so in a new, novel way.

I am surprised by the use of “uniform global attractor property” rather than “indexability” throughout the paper. With regards to clarity, I encourage the authors to include more explanations to inform the reader.

The contributions of this paper are purely theoretical. Open-source code as well as empirical results comparing FTVA against other state-of-the-art implementations would be helpful for the research community.

[1] Abheek Ghosh, Dheeraj Nagaraj, Manish Jain, and Milind Tambe. 2023. Indexability is Not Enough for Whittle: Improved, Near-Optimal Algorithms for Restless Bandits. In Proceedings of the 2023 International Conference on Autonomous Agents and Multiagent Systems (AAMAS '23). International Foundation for Autonomous Agents and Multiagent Systems, Richland, SC, 1294–1302.


**Questions:**

Are the authors aware of [1]? That paper also claims to contribute a novel new approach for the discrete-time problem. Not only do they consider average-reward, but also any discounted total expected reward in both the finite and infinite horizon case. How does your approach differ? How do your optimality guarantees differ?

[1] Abheek Ghosh, Dheeraj Nagaraj, Manish Jain, and Milind Tambe. 2023. Indexability is Not Enough for Whittle: Improved, Near-Optimal Algorithms for Restless Bandits. In Proceedings of the 2023 International Conference on Autonomous Agents and Multiagent Systems (AAMAS '23). International Foundation for Autonomous Agents and Multiagent Systems, Richland, SC, 1294–1302.


**Limitations:**

Yes.

---

> ### Author Rebuttal · Authors · 2023-08-09
>
> We thank the reviewer for recognizing our paper’s contribution and strengths, and for the constructive comments. We provide our detailed responses below.
>
> > Relation with Ghosh et al. (AAMAS 23).
>
> We thank the reviewer for pointing out this related work. We address this point in detail in our **global rebuttal**. We provide a brief summary here: Our work differs from Ghosh et al. (AAMAS 23) in terms of the problem setting (finite-horizon or discounted-reward in Ghosh et al, and infinite-horizon average-reward in our paper) and the algorithmic approach; moreover, significant challenges arise when one attempts to apply the techniques from Ghosh et al to our setting. We will add this discussion in the revised paper.
>
> > Uniform global attractor property (UGAP) v.s. indexability
>
> Thanks for raising this point. There is indeed a nuanced distinction between UGAP and indexability, which should be made more explicit in the paper. Indexability is a condition specific to the Whittle Index Policy (WIP), but indexability alone is not enough to guarantee the asymptotic optimality of WIP [WW90]. In contrast, UGAP has been a condition imposed by prior work to prove the asymptotic optimality and bound the optimality gap of both WIP and the LP-Priority policy [WW90, Ver16, GGY20, GGY21]. Given that work on LP-Priority policy [Ver16,GGY22] has removed the reliance on indexability, we believe that UGAP is a more fundamental condition to existing policies. Therefore, we have focused on breaking UGAP.
>
> We acknowledge the need for clearer differentiation between UGAP and Indexability, and we will include this explanation in our revision for better clarity.
>
> > Open-source code and empirical results
>
> Thanks for this suggestion. We will make our source code and the empirical results available on GitHub. Our source code includes implementations of our proposed policy FTVA and two state-of-the-art policies, the Whittle Index Policy and LP-Priority policy.

---

### Official Review · Reviewer_TkxD · 2023-07-05

**Soundness:** 3 good
**Presentation:** 3 good
**Contribution:** 3 good
**Rating:** 6
**Confidence:** 4

**Summary:**

This paper studied, for both the discrete-time and continuous-time settings, the average-reward restless bandit problem without the uniform global attractor property. They proposed the simulation-based policies Follow-the-Virtual-Advice (FTVA) and FTVA-CT, which converts the optimal single-armed policy into the N-armed constrained policy by simulating a virtual process that follows the single-armed policy, and then by letting the real process act closely to the virtual process but satisfying the constraints. The policies provide vanishing performance loss.

**Strengths:**

Novelties: In this reviewer's knowledge, this paper is the first paper to provide an algorithm for the average reward constraint RB problem without uniform global attractor property, which is a property that is hard to verify. The proposed FTVA policy is quite straightforward and also easy to understand. The idea of using virtual processes in this problem seems original to me.

Presentation: The paper is well-written, with precise language and clear structure. The paper is clear and easy to understand. And there are a lot of intuitive explanations.

Significance: The paper provides the first algorithm for the average reward constraint RB problem without uniform global attractor property.

**Weaknesses:**

The following assumption is very strong and important to this policy: the author assumed that all the arms with the same state and action have the same reward. And this policy seems very hard to extend to the case where the arms have different reward functions. (More comments in Questions.)

The paper feels incomplete, which prevents a clear conclusion at the end.

The simulation is not sufficient, the author only showed the average reward comparison for different policies, but I would also want to know how is the action look like for different policies or is there any intuition in it.

The paper presentation is too high-level and has missing details. The author claimed at first, but didn’t explain later, how the FTVA policy can be implemented in a distributed manner. The author claimed the algorithm can run at a linear computational cost but didn’t explain how, later.

**Questions:**

In the setup, the author assumed that all the arms with the same state and action have the same reward. With this assumption, the problem (with the \alpha N constraint) is greatly simplified. This assumption makes me feel that even with the \alpha N constraint, those arms are still acting very independently, not affecting each other too much. And this policy seems very hard to extend to the case where the arms have different reward functions.

If possible, instead of only the average reward comparison, I am also curious about how the FTVA action policy look like compared to other non-asymptotically optimal policies in the example. And is there any significant different actions on some special states that makes the policy be optimal?

In the beginning, the author said that FTVA policy can be implemented in a distributed manner for arms, could you add more about it after the algorithm descriptions?

Figure1 is missing one label.

**Limitations:**

There is no negative societal impact. Discussion on limitations is not extensive.

---

> ### Author Rebuttal · Authors · 2023-08-10
>
> We thank the reviewer for the constructive comments. We recap the comments and present our responses below. We will revise our paper accordingly.
>
> > Assumption of homogeneous arms.
>
> We address this comment in the global rebuttal, as this is an important point. In a nutshell, our approach and results can be directly generalized to accommodate heterogeneous arms in a broad class of settings.
>
> > Lack of conclusion.
>
> We will add a conclusion in the revision.
>
> > Simulations that demonstrate actions.
>
> We thank the reviewer for this suggestion. We produce more illustrations for the example in Section 3.3 of our paper, which can be found in the pdf attached to the global rebuttal. In this example, we can tell that a policy is good if the fraction of arms in each state is roughly uniform, and bad if the arms concentrate on a small number of states. We can also see why this happens by looking at whether the preferred action is chosen, and whether the arms are moving from $s$ to $(s+1) \text{ mod } 8$.
>
> The three heatmaps in Figure R2(a) illustrate how the fractions of arms in each state change over time under the three policies: random tie-breaking, LP-Priority, and FTVA. We can see that the under the random tie-breaking and LP, the arms concentrate around a few states, whereas under FTVA, the arms are uniformly spread out over the 8 states.
>
> In Figure R2(b), we take a closer look at the sample path under the random tie-breaking policy from time $250$ to $289$ and contrast it with the sample path if the system uses FTVA from time $250$ onwards. We also add arrows indicating the *drift* of the arms, i.e., the average direction that arms in this state are moving in. The colors of the arrows represent the direction: blue implies that the arm takes the preferred action and moves in the correct direction, while red implies that the arm takes the non-preferred action and moves in the wrong direction.  The magnitudes of the arrows represent moving speeds.
>
> We can see that under random tie-breaking, although the arms have a strong tendency to move up from state $0$ to state $1$, it moves back when it reaches state $1$ and thus get stuck at state $0$. In contrast, when using FTVA, most arrows point in the right direction, which implies that the arms consistently apply the preferred actions. As a result, FTVA helps the arms to escape from state $0$ and converge to the uniform distribution over the state space.
>
> Figure R2(d) illustrates the fraction of arms in state $s$ taking action $a$ in $4$ time steps post time $250$. The upper and lower bar plots represent the random tie-breaking policy and FTVA, respectively. The x-axis denotes the state; the y-axis denotes the fraction; each state has $4$ bars for the $4$ time steps. Each bar has at most two segments, blue indicates the fraction for the preferred action, and red for the non-preferred.
>
> We can see that under FTVA, more arms choose the preferred actions for state $1$ than under the random tie-breaking policy, which prevents the arms in state $1$ to move back to state $0$ as we see in Figure R2(b) under the random tie-breaking policy.
>
> Figures R2(c) and R2(e) demonstrate a similar comparison between LP-Priority and FTVA.
>
> The common failure mode for random tie-breaking and LP-Priority in this example is that the arms concentrate on a bad state $s$, which prevents the arms on state $s+1$ from applying the preferred action due to the budget constraint. This causes a *livelock* where the arms move back and forth between state $s$ and $s+1$, and fail to gain a reward by moving past state $7$. FTVA solves this issue by letting more arms in state $s+1$ follow the preferred action, which breaks the livelock and helps all arms converge to the optimal distribution.
>
> > The paper presentation is too high-level and has missing details.
>
> We appreciate the reviewer’s comments here as they help us improve our presentation.  We address the reviewer’s concerns on the implementation and the computational cost of FTVA below.  We are open to any further suggestions.
> - > How will the FTVA policy be implemented in a distributed manner?
>
>     We appreciate this question.  This is indeed a point that we would like to make more precise.  In the FTVA policy, each arm simulates a virtual single-armed process and generates a virtual action in each time step.  Since the virtual processes are independent across arms, these simulations can be implemented in a fully distributed manner.  When taking the real actions, the policy needs to know the virtual actions from all arms.  So this component cannot be distributed.  We will make sure to clarify this in the revision.
>
> - > Why does the algorithm have a linear computational cost?
>
>     We will also add the following explanation for the computational cost of FTVA after the policy description.
>
>     To see why the computational cost of the FTVA policy is linear in N, we examine the cost component-by-component.
>
>     - Firstly, FTVA needs to solve the LP in (5)-(7) once.  Given that this LP doesn’t depend on N, the computational cost here is a constant.
>     - Secondly, each arm simulates a virtual process.  As the computation required for each simulation doesn’t scale with N, the overall computational cost here is linear in N.
>     - Lastly, in each time step, FTVA selects $\alpha N$ arms from a subset of arms to determine the real actions.  The computational cost for this selection process scales linearly with N.
>
>     Combining these components, we can see that the computational cost of FTVA is linear in N.
>
> > Figure 1 missing one label.
>
> We believe that the reviewer refers to the line below the curve for Whittle Index.  That line is meant to indicate where the award under Whittle Index converges to as N becomes large.  We will remove it to avoid confusion.

---

> > ### Comment · Reviewer_TkxD · 2023-08-19
> >
> > I thank the authors for a thorough and satisfactory response. I will update my rating.

---

### Official Review · Reviewer_k12y · 2023-07-07

**Soundness:** 3 good
**Presentation:** 3 good
**Contribution:** 3 good
**Rating:** 8
**Confidence:** 2

**Summary:**

This work studies the restless bandit problem. In this setting the agent controls a set of $N$ arms modelled as Markov Decision Processes with two actions (active, inactive). The MDPs are coupled with a constraint that forces the agent to maintain a fixed arm activation rate. The authors propose a novel algorithm for this problem that does not use the commonly used uniform global attractor property (UGAP). The key insight of the approach is to transform a learned single arm policy to a $N$ armed policy. While this may lead to a discrepancy between the learned policy and what is actually played because of the constraint, the proposed algorithm aims to bring these two states closer together. The authors provide convergence guarantees and rates for both discrete and continuous time variants of the algorithm.

**Strengths:**

To the best of my knowledge, the proposed algorithm is novel. The authors also presented very clearly the setup of the problem which is especially important given that many members of the community may not be familiar. While I did not follow all the technical details in this work, the synchronization assumption seems like a significantly weaker assumption than UGAP, indicating this a very strong contribution. On top of that, the way the algorithm solves the problem is itself highly non trivial.

**Weaknesses:**

I think clarifying a little bit more the importance of synchronization when we are using the constraint would be very helpful. I will try to outline my understanding in case it is helpful in improving the presentation:

* In Section 4.1, based on the stated assumptions it is clear that achieving synchronization once is sufficient to achieve synchronization forever. But for the restless bandit setting it seems that the real and virtual states synchronize and desynchronize in a loop. Indeed the states start synchronized and some arms may immediately get desynchronized (become bad in the terminology of Theorem 1).
* I understand that the the SA assumption helps us understand desynchronization is resolved in finite time. But if synchronization lasts only a single round, then the arms will still be bad most of the time. So SA alone is not sufficient.
* Based on the proof sketch of Theorem 1, it seems like desynchronization is inherently rare for large $N$ so this is why SA is sufficient. Intuitively, SA is only needed to bound the effect of a single good-bad-good arm loop.

If my understanding is indeed correct, it would be helpful to make this more explicit somewhere in the text, preferably in the context of the example of Section 3.3.

**Questions:**

See above for some suggestions.

**Limitations:**

All limitations are addressed.

---

> ### Author Rebuttal · Authors · 2023-08-09
>
> We thank the reviewer for appreciating our work, and for the suggestion on further clarifying the role and importance of SA. Your understanding is well aligned with the intended role of SA and with our proofs. We will integrate your suggestion into Section 3.3, specifically within lines 185-193 and in the example.

---

> > ### Comment · Reviewer_k12y · 2023-08-19
> > **Thanks for the clarification**
> >
> > I would like to thank the authors for the clarification!

---

### Official Review · Reviewer_JXov · 2023-07-27

**Soundness:** 3 good
**Presentation:** 4 excellent
**Contribution:** 3 good
**Rating:** 6
**Confidence:** 2

**Summary:**

This paper studies the problem of optimizing a policy for restless bandits in the infinite horizon case. A new algorithm is proposed that can convert a single-arm policy into an N-arm policy using virtual trajectories of the states and actions of the single-arm policy. The main idea is to generate virtual actions using the single-arm policy and then attempt to match these virtual actions with real actions that match the budget constraint. The algorithm is shown to have suboptimality that degrades with N. Both discrete time and continuous time versions are given. One of the main contributions is that this paper forgoes the UGAP condition.

This is an emergency review. I am not familiar enough with this literature to definitively judge the significance and novelty.

**Strengths:**

The paper appears to tackle an important problem in the literature broadly (restless bandits) and provides a potential solution to optimize policies with vanishing suboptimality as N increases.

Achieving this without the UGAP condition that appears in prior work is interesting.

Overall the paper is well written and generally easy to understand. The examples and organization of the paper help convey these results.

**Weaknesses:**

The main weakness appears to be that the entire premise hinges on the synchronization assumption that the real states and virtual states will eventually match, if the same actions are played, after which point they will remain identical. This seems reasonable in some settings, but a more clear contrast with conditions in other papers would be helpful.

In general, it would be helpful to better understand the significance of this change of assumptions and how this might translate into practice.

**Questions:**

It is mentioned that Algorithm 1 should “steer” the trajectories towards the virtual trajectories but still respect the budget constraints. However, in Line 4, the selection of the smaller set to respect the budget does not appear to filter arms in any particular way. What prevents the same arms from being activated while starving others and thus causing the virtual trajectory to differ from the real one by a lot?

Suggestion: the paper could benefit from further discussion of the last theorem and a conclusion.

**Limitations:**

No negative societal impact, but it would be helpful to have more discussion of limitations.

---

> ### Author Rebuttal · Authors · 2023-08-09
>
> We thank the reviewer for the constructive comments.  Below we quote the reviewer's key comments and present our responses.
>
> > Compare the synchronization assumption (SA) with conditions in other papers, and better explain the significance of SA.
>
> Before presenting the comparison, we reiterate that for the continuous-time setting, our work *does not require SA*. Therefore, we focus on the discrete-time setting.
>
> All prior work on discrete-time, infinite-horizon average-reward RBs relied on UGAP. We hence contrast SA and UGAP below. Most of the content below is in the paper, but we will make sure to better highlight the contrast and improve the exposition.
>
> 1. **There exist problem instances where SA holds but UGAP does not.**
> It is unclear whether SA subsumes UGAP. This is recognized (in lines 61-62) as a limitation of our work. However, there exist problem instances where SA holds but UGAP doesn’t. Two such examples are given in our paper (illustrated in Figures 1 and 2), where the first example is borrowed from [GGY20]. The work [GGY20] provides more examples that do not satisfy UGAP. One can easily see that these examples satisfy SA: all entries in the two transition matrices therein are positive, so two arms starting from any pair of initial states have a positive probability of synchronizing in one step (a formal argument is given in Proposition 2 in Appendix A). For these examples, FTVA achieves an $O(1/\sqrt{N})$ optimality gap; previously, no policy was known to be asymptotically optimal. Another perspective to this comparison is that SA uncovers the feasibility of achieving asymptotic optimality for more problems.
>
>     &nbsp;&nbsp;&nbsp;&nbsp; Another helpful comparison is the following. For any problem instance, we can make it satisfy SA by perturbing entries of its transition kernels by an arbitrarily small amount such that all the entries are strictly positive. This implies that the set of problem instances satisfying SA is dense in the problem space. In contrast, UGAP lacks this denseness property. In particular, UGAP assumes the existence of a global attractor. It can be shown that the set of instances that do not satisfy UGAP contains open sets.
>
> 2. **SA is easier to verify.**
>
>     It has been well recognized in many prior papers that UGAP is tricky to verify. It lacks intuitive sufficient conditions. Prior work has assessed UGAP through numerical experiments, where one simulates the mean-field ODE with randomly generated initial states. However, since not all initial states can be covered, this method can confirm *violation* of UGAP but it cannot certify compliance of UGAP.
>
>    &nbsp;&nbsp;&nbsp;&nbsp; In contrast, SA can be efficiently verified in several ways. By definition, SA involves the hitting time for the finite-state Markov chain $(S(t), \widehat{S}(t))$ (eq. (10)) to reach the subset of states $\lbrace (s,\hat{s})\colon s=\hat{s} \rbrace$. Therefore, SA can be efficiently verified using path-finding graph algorithms on the state transition diagram of the Markov chain. Additionally, we provide several intuitive sufficient conditions for SA in Appendix A, which can also be efficiently verified by identifying self-loops and cycles in the transition diagram.
>
> > Can Algorithm 1 activate the same arms while starving others?
>
> Algorithm 1 will not activate the same arms while starving others for a significant amount of time.
>
> Firstly, it is important to observe that the set of arms $\mathcal{A}$ activated in Line 4 of Algorithm 1 does not significantly deviate from the larger set, $\lbrace i\colon \widehat{A}_i(t) = 1 \rbrace$.  Indeed, we’ve shown that the cardinality of $\lbrace i\colon \widehat{A}_i(t) = 1\rbrace$ is $\alpha N \pm O(\sqrt{N})$ (precise statement in Lemma 2 in Appendix B.2).
>
> We therefore only need to focus on the composition of the set $\lbrace i\colon \widehat{A}_i(t) = 1\rbrace$, which is a random set that varies from time slot to time slot.  Specifically, for each arm $i$, the virtual action $\widehat{A}_i(t)$ is generated according to the solution of the LP in (5)-(7).  The constraints in the LP ensure that the arm generates $\widehat{A}_i(t)=1$ for an $\alpha<1$ fraction of time.  Therefore, each given arm stays in the set $\lbrace i\colon \widehat{A}_i(t) = 1\rbrace$ for an $\alpha<1$ fraction of time.  This has two implications:
>
> - The algorithm will not consistently activate the same set of arms:  the set $\lbrace i\colon \widehat{A}_i(t) = 1\rbrace$ cannot comprise a fixed set of arms since any given arm will leave the set when its virtual action is 0, which happens for a $1-\alpha$ fraction of time.
>
> - No arm will be starved: the set $\lbrace i\colon \widehat{A}_i(t) = 1\rbrace$ will cycle through all arms, since each arm generates a virtual action of 1 for an $\alpha$ fraction of time.
>
> > Suggestion for adding further discussion of the last theorem and a conclusion.
>
> Thanks for the suggestion.  We will add the suggested discussion, highlighting the connection and differences compared with the discrete-time setting, and a conclusion.

---

> > ### Comment · Reviewer_JXov · 2023-08-19
> > **Reply**
> >
> > Thank you for the thorough response. I will update my score.

---

### Author Rebuttal · Authors · 2023-08-09

We thank all the reviewers for their time and effort. In this global rebuttal, we address two points each raised by multiple reviewers. Our responses to each reviewer are provided in the individual rebuttals.

## Comparison with prior work by Ghosh et al. (@Reviewers s6TA, S8QK, TTwr)
We thank Reviewers s6TA and TTwr for pointing us to recent work by Ghosh et al. (ArXiv 2022, AAMAS 2023), referred to as [Ghosh23] below. We apologize for the omission and will add it as related work in our revision. A quick summary of the comparison between [Ghosh23] and our work is that: the settings differ (finite-horizon or discounted-reward vs infinite-horizon average-reward), the algorithmic approaches differ, and a crucial obstacle emerges if one tries to apply the techniques in [Ghosh23] to our setting. Below we elaborate on this comparison; comparison with some other prior work on finite-horizon/discounted-reward [ZF21, ZF22] is also mentioned along the way.

### – Settings
[Ghosh23] focuses on the finite-horizon setting and the infinite-horizon discounted-reward setting, while our work considers the infinite-horizon average-reward setting.

### – Approaches
- While linear programs (LPs) are used in policy design in both the finite-horizon/discounted-reward setting (including [Ghosh23], [ZF21] and [ZF22]) and the infinite-horizon average-reward setting (including our work), the LPs utilized differ substantially. In the finite-horizon/discounted-reward setting, the LP optimizes the future trajectory for a predetermined number of time steps, hence the number of optimization variables scales linearly with the time steps. In contrast, in the infinite-horizon average-reward setting, the LP solves for the optimal state-action frequency in steady state, effectively removing any consideration of the number of time steps.
- [Ghosh23] directly takes actions according to the LP solution, while we simulate a virtual process and generate actions using the virtual process. We reiterate that the virtual process is a novel approach that plays a crucial role in breaking the UGAP assumption.

### – Obstacle
In general, the infinite-horizon average-reward setting faces difficulties due to the long-term effect of actions, whereas the finite-horizon and discounted-reward settings have bounded (effective) horizons. Standard results on MDPs show that one can use the discounted reward to approximate the infinite-horizon average reward by dividing the former by the effective horizon $1/(1-\gamma)$ and letting $\gamma\to 1$. Therefore, for an optimality gap bound in the discounted-reward setting to be useful for average reward, it should have at most linear dependence on $1/(1-\gamma)$. However, the bound in Theorem 3 of [Ghosh23] has a *quadratic* dependency on $1/(1-\gamma)$. In the finite-horizon setting, the bound on the additive optimality gap in Theorem 2 of [Ghosh23] also has a *quadratic* dependency on the time horizon T.

We believe that new techniques are required to improve the quadratic dependency on  $1/(1-\gamma)$ or T to a linear dependency. Our intuition is that a certain form of fast-mixing property of the system must be utilized. UGAP is one such fast-mixing property that has been assumed for 30 years. Our work takes an alternative approach by utilizing the fast mixing of real and virtual states.

## Assumption of homogeneous arms and extension to heterogeneous arms (@Reviewers TkxD, TTwr)
Reviewers TkxD and TTwr raised concerns on our assumption of homogeneous arms and applicability to settings with heterogeneous arms. We appreciate this point, since we too think that the heterogeneous setting is interesting. Our paper has focused on breaking the UGAP assumption for the infinite-horizon average-reward setting, for which the homogeneous-arm setting is used as an exemplary case. Our approach and results can be directly generalized to accommodate heterogeneous arms as follows.

Suppose the arms are divided into $K$ types, with $\beta_k N$ arms in each type $k\in\{1,2,\dots,K\}\triangleq[K]$, and each type is associated with an MDP. The transition kernels and reward functions can be different across types. This is similar to the clustering setting in [Ghosh et al., AAMAS 2023]. When $K$ is a constant as $N$ increases, we generalize our approach as follows. The objective of the LP in (5)-(7) now becomes

$\text{maximize}\sum_{k=1}^K\sum_{s,a}\beta_kr_k(s,a)y_k(s,a)$

with variables $\lbrace y_k(s,a)\rbrace_{{k\in [K]},s\in\mathbb{S},a\in\mathbb{A}}$.

The constraint (5) becomes $\sum_{k=1}^K \beta_ky_k(s,1)=\alpha.$

The constraints (6) and (7) now need to hold for each $k$. Then the solution, $y_k(s,a)$, induces an optimal single-armed policy $\pi_k^*$ for an arm of type $k$. To convert from single-armed policies to an $N$-armed policy, FTVA lets each arm of type $k$ independently simulate a virtual single-armed process following $\pi_k^*$. Lines 3-14 of Algorithm 1 stay the same.

For this heterogenous setting, assuming the Synchronization Assumption for each arm type, the optimality gap remains $O(1/\sqrt{N})$, with the hidden constant depending on $K$. Except for the adaptation to incorporate multiple arm types, the proof structures are almost the same, with only minor changes needed. We will include this generalization in the appendix in our revision.

We acknowledge that in the setting where $K$ scales with $N$, or with multi-actions, more significant adaptations might be required. Nonetheless, we believe that our ideas of simulation-based policy design and prioritizing constraint satisfaction remain valid, and it is plausible that our FTVA approach will still be applicable.

---

### Decision · Program_Chairs · 2023-09-21

**Decision:**

Accept (spotlight)

**Comment:**

This paper studies a key scalability bottleneck in multi-armed bandit problems with discrete but countably large number of actions, and identifies a way to reduce solving such problems to a scalable subset through an algorithm called Follow-the-Virtual-Advice. Performance bounds quantify the gap between the reduction and the original problem in terms of regret. During the discussion period, the authors addressed the majority of the reviewer concerns, and came to the conclusion that this work contributes an important development to the theory of multi-armed bandits. I recommend it be accepted.